# Distinct but overlapping roles of LRRTM1 and LRRTM2 in developing and mature hippocampal circuits

**Shreya H Dhume[1,2], Steven A Connor[3,4], Fergil Mills[5], Parisa Karimi Tari[3,4], Sarah HM Au-Yeung[3], Benjamin Karimi[1,2], Shinichiro Oku[1,2,3], Reiko T Roppongi[1,2], Hiroshi Kawabe[6,7,8,9], Shernaz X Bamji[5], Yu Tian Wang[10], Nils Brose[6], Michael F Jackson[11], Ann Marie Craig[3]\*, Tabrez J Siddiqui[1,2,12,13]\***

[1]Neuroscience Research Program, Kleysen Institute for Advanced Medicine, Health Sciences Centre, Winnipeg, Canada; [2]Department of Physiology and Pathophysiology, University of Manitoba, Winnipeg, Canada; [3]Department of Psychiatry and Djavad Mowafaghian Centre for Brain Health, University of British Columbia, Vancouver, Canada; [4]Department of Biology, York University, Toronto, Canada; [5]Department of Cellular and Physiological Sciences, University of British Columbia, Vancouver, Canada; [6]Department of Molecular Neurobiology, Max Planck Institute for Multidisciplinary Sciences, Göttingen, Germany; [7]Division of Pathogenetic Signaling, Department of Biochemistry and Molecular Biology, Kobe University Graduate School of Medicine, Kobe, Japan; [8]Department of Gerontology, Laboratory of Molecular Life Science, Institute of Biomedical Research and Innovation, Foundation for Biomedical Research and Innovation at Kobe, Kobe, Japan; [9]Department of Pharmacology, Gunma University Graduate School of Medicine, Gunma, Japan; [10]Division of Neurology, Department of Medicine and Djavad Mowafaghian Centre for Brain Health, University of British Columbia, Vancouver, Canada; [11]Department of Pharmacology and Therapeutics, University of Manitoba, Winnipeg, Canada; [12]The Children's Hospital Research Institute of Manitoba, Winnipeg, Canada; [13]Program in Biomedical Engineering, University of Manitoba, Winnipeg, Canada

**\*For correspondence:** acraig@mail.ubc.ca (AMC); tabrez.siddiqui@umanitoba.ca (TJS)

**Abstract** LRRTMs are postsynaptic cell adhesion proteins that have region-restricted expression in the brain. To determine their role in the molecular organization of synapses in vivo, we studied synapse development and plasticity in hippocampal neuronal circuits in mice lacking both *Lrrtm1* and *Lrrtm2*. We found that LRRTM1 and LRRTM2 regulate the density and morphological integrity of excitatory synapses on CA1 pyramidal neurons in the developing brain but are not essential for these roles in the mature circuit. Further, they are required for long-term-potentiation in the CA3-CA1 pathway and the dentate gyrus, and for enduring fear memory in both the developing and mature brain. Our data show that LRRTM1 and LRRTM2 regulate synapse development and function in a cell-type and developmental-stage-specific manner, and thereby contribute to the fine-tuning of hippocampal circuit connectivity and plasticity.

## Editor's evaluation

This study aims to systematically investigate the role of LRRTM 1 and 2 in hippocampal circuit function. These neuronal proteins are organizing proteins in the synapse and are known to play a role

in development and plasticity. The current literature is unclear on the exact role these proteins play due to the diverse technical approaches used in previous studies which often led to contradictory findings. The current study aims to clearly identify the roles of these proteins by using a thorough experimental design. The clear strength of the study is in the systematic and careful comparison between various approaches that aim to eliminate LRRTM1/2 from the synapses and the thorough examination of several parameters under these different conditions.

## Introduction

Neurons are interconnected by specialized and highly differentiated synaptic junctions. Synapse organizing proteins coordinate the precise alignment of pre- and post-synaptic components of a synapse and impart them with distinct physiological properties necessary for neuronal circuits to function and control behavior. Presynaptic neurexins, arguably the best characterized synapse organizers, bind to multiple postsynaptic ligands, including neuroligins and leucine-rich-repeat transmembrane neuronal proteins (LRRTMs) (*Südhof, 2017*), and act as hubs for presynaptic development and signaling (*Reissner et al., 2013*; *Südhof, 2017*).

The LRRTMs are a family of four postsynaptic transmembrane proteins that instruct presynaptic differentiation in contacting axons, and mediate postsynaptic recruitment of scaffold proteins and transmitter receptors (*Linhoff et al., 2009*; *Roppongi et al., 2017*). In cell culture studies, LRRTM1 and LRRTM2 exhibited similar synaptogenic activities through dual binding to the heparan sulfate modification and splice variants of neurexins (*Roppongi et al., 2020*; *Zhang et al., 2018*) indicating that LRRTM1 and LRRTM2 likely have overlapping functions. In contrast to neuroligins, which have broad expression profiles (*Varoqueaux et al., 2006*), individual LRRTMs are expressed in a region-restricted manner in the brain (*Laurén et al., 2003*). Within the hippocampus, for instance, LRRTM1 and LRRTM2 are the only family members expressed in the CA1 region ,whereas all LRRTMs are expressed in the dentate gyrus (DG).

Accordingly, LRRTMs are thought to function as cell-type selective synapse organizers, but the exact roles of LRRTM1 and LRRTM2 in synapse development have remained unresolved as corresponding studies yielded divergent and partially contradictory results. ShRNA-based knock-down (KD) studies in cultured neurons or brain slices produced differing results (*de Wit et al., 2009*; *Ko et al., 2011*; *Soler-Llavina et al., 2011*), where KD of LRRTM2 alone reduced the numbers of excitatory synapses (*de Wit et al., 2009*) whereas combined KD of LRRTM1 and LRRTM2 did not affect synapse numbers (*Ko et al., 2011*). Further, KD or genetic knock-out (KO) of *Lrrtm1* in mice causes a~10% loss of excitatory synapses and a redistribution of synaptic vesicles in certain sublaminae of the hippocampal CA1 region (*Linhoff et al., 2009*; *Schroeder et al., 2018*; *Takashima et al., 2011*). These subtle alterations are likely due to functional compensation by LRRTM2, which is co-expressed with LRRTM1 in the CA1 region. Finally, it is unclear whether the synapse loss caused by *Lrrtm1* deletion is due to a role of LRRTM1 in synapse formation or the prevention of their elimination. While the reasons for the multiple discrepant data on LRRTM1 and LRRTM2 are not known, it is likely that the well-known confounds of KD approaches, overexpression artifacts, and ectopic synaptogenic activity have played a major role in in vitro studies but less so in experiments involving in vivo models (*Südhof, 2018*).

Due to their role in synapse formation, maturation, and function, LRRTM1 and LRRTM2 are critical players in multiple aspects of circuit and brain function. Most notably, LRRTM1 and LRRTM2 contribute to long-term potentiation (LTP) in CA1 pyramidal neurons. This is likely due to their ability to anchor AMPA receptors at synapses (*Bhouri et al., 2018*; *Ramsey et al., 2021*), but what is not known in this context is whether LRRTMs are required for LTP in other cell-types that express them. This is an important open question because synaptic plasticity is a partially cell-type specific phenomenon (*Larsen and Sjöström, 2015*). In human, alterations in *LRRTM* genes are associated with several neuropsychiatric disorders, including autism, schizophrenia, and bipolar disorder (*Francks et al., 2007*; *Leach et al., 2014*; *Ludwig et al., 2009*; *Malhotra et al., 2011*; *Pinto et al., 2010*), and in partial accord with human genetics, constitutive or conditional *Lrrtm1* KO mice exhibit defined forms of cognitive impairment (*Karimi et al., 2021*; *Takashima et al., 2011*; *Voikar et al., 2013*). However, it is unclear what neural circuit origins these dysfunctions have or whether the contribution of LRRTM1 and LRRTM2 to cognitive processes stems from their role in developmental synapse organization or synaptic plasticity.

In the present study, we devised spatially and temporally controlled genetic approaches in mice to investigate the contributions of LRRTM1 and LRRTM2 to synapse organization, synaptic plasticity, and cognitive function in the developing and mature brain. We assessed the joint roles of LRRTM1 and LRRTM2 in the hippocampal CA1 region, where LRRTM3 and LRRTM4 are not expressed, and compared their role in the dentate gyrus, where all four LRRTMs are co-expressed. Our results reveal cell-type and development-stage dependent functions of these LRRTMs in synapse development and synaptic transmission, but broad roles in synaptic plasticity and cognitive function, indicating a context-dependent role for these synapse organizers in the molecular organization of neural circuits.

## Results

### Basic characteristics of constitutive *Lrrtm1* and *Lrrtm2* (LRRTM1/2) double KO (DKO) mice

To investigate the role of LRRTM1 and LRRTM2 in developing neural circuits, we generated constitutive LRRTM1/2-DKO mice. Briefly, the major coding exon 2 was excised in both *Lrrtm1* and *Lrrtm2* by crossing the *Lrrtm1*^floxed/floxed^ and *Lrrtm2*^floxed/floxed^ mice (*Bhouri et al., 2018*) with the EIIa-Cre line (*Lakso et al., 1996*; *Figure 1—figure supplement 1*). The heterozygous mice thus generated were crossed to obtain homozygous KO mice. Single KOs were then crossed to obtain LRRTM1/2-DKOs. The expected complete loss of LRRTM1 and LRRTM2 was confirmed by western blotting of brain homogenates (*Figure 1—figure supplement 1*). Studies were performed on DKO mice and control wild-type (WT) mice, derived from the same double heterozygous parents and matched for age and sex.

LRRTM1/2-DKOs survived and bred normally at the predicted Mendelian frequencies, had normal body weight, and exhibited no obvious behavioral abnormalities in the home cage environment. Further, LRRTM1/2-DKOs were indistinguishable from WT mice with respect to gross brain morphology and cytoarchitectural organization as assessed by confocal microscopy analysis of brain sections labeled for the nuclear marker DAPI and the synaptic markers vesicular glutamate transporter (VGlut1) and glutamic acid decarboxylase (GAD65) (*Figure 1A*). To determine whether embryonic loss of *Lrrtm1* and *Lrrtm2* alters synaptic composition globally in the brain, we prepared crude synaptosomal fractions from whole brain of LRRTM1/2-DKO and control WT mice at 6–7 weeks postnatally (*Figure 1—figure supplement 2*). Quantitative immunoblotting of these fractions revealed no differences in the levels of the active zone proteins bassoon and CASK, the SNARE proteins SNAP-25 and VAMP2, the synaptic vesicle markers VGlut1 and synapsin, the inhibitory synapse scaffolding protein gephyrin, the excitatory synapse scaffolding proteins PSD95, SAPAP, SHANK, and SynGAP, the ionotropic glutamate receptor subunits GluA2, GluN1, and GluN2A, and the adhesion/receptor proteins neuroligin 1, neuroligin 2, LRRTM4, and TrkC. A small but significant increase in the level of the synaptic vesicle protein synaptophysin was observed in LRRTM1/2-DKO synaptosomes. These results indicate that loss of LRRTM1 and LRRTM2 does not substantially affect global synaptic protein content.

### LRRTM1 and LRRTM2 control the numbers of excitatory synapses and spines selectively in CA1

In the assays described above, changes in the protein composition of synapses in specific brain regions might have gone unnoticed due to contributions from unaffected brain regions. We therefore focused further analyses on specific circuits, comparing the hippocampal CA1 region with the dentate gyrus. To this end, we performed high-resolution confocal imaging of markers of excitatory and inhibitory synapses in CA1 and dentate gyrus dendritic layers, at 6–7 weeks postnatally. As compared to control WT mice, quantitative confocal analysis of LRRTM1/2-DKOs revealed significantly reduced VGlut1 puncta immunofluorescence in all CA1 dendritic layers, that is stratum oriens, stratum radiatum, and stratum lacunosum (*Figure 1B and C*). In the dentate gyrus, however, VGlut1 puncta immunofluorescence was elevated in the outer molecular layer but remained unchanged in the inner and medial molecular layers of LRRTM1/2-DKOs when compared to controls (*Figure 1D and E*), indicating potential compensation by other synapse organizers in the dentate gyrus of LRRTM1/2-DKOs. Puncta immunofluorescence of GAD65 in the same co-labeled regions in both the CA1 and dentate gyrus was comparable between LRRTM1/2-DKO and control mice (*Figure 1—figure supplement 3*). Thus,

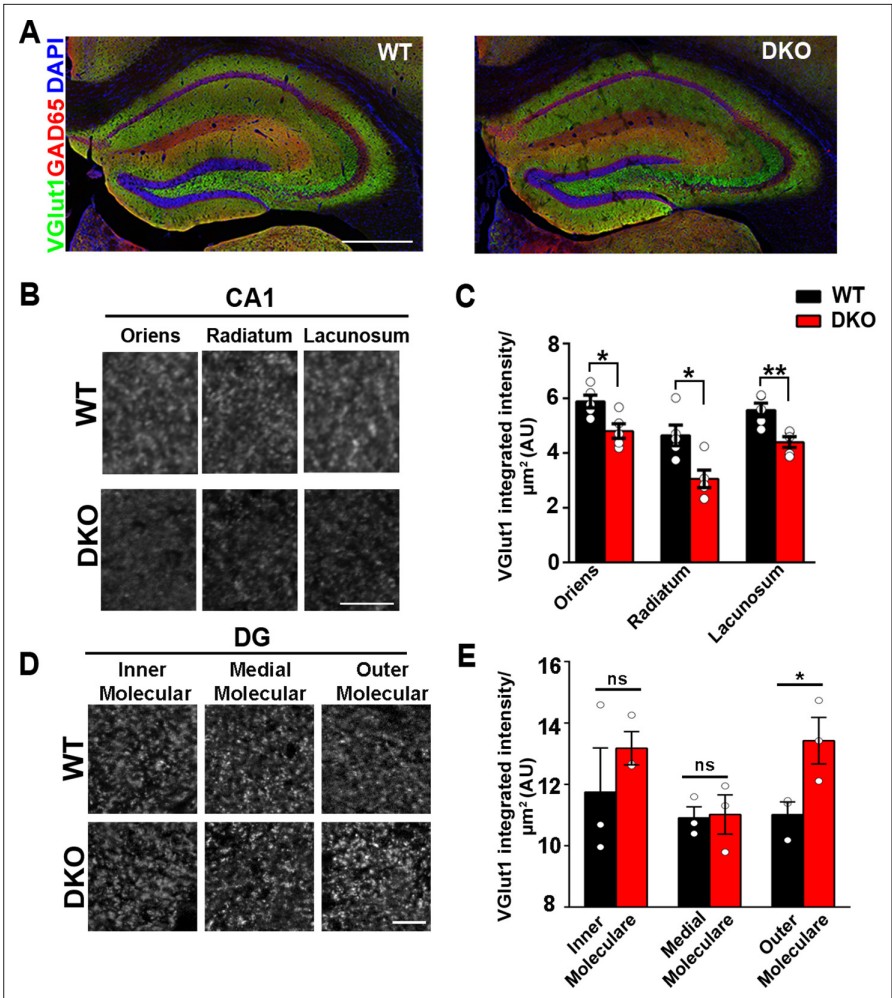

**Figure 1.** LRRTM1 and LRRTM2 control excitatory presynapse development in the hippocampus. (**A**) Confocal immunofluorescence images for VGlut1, GAD65 and the nuclear marker DAPI revealed normal hippocampal morphology and large-scale synaptic organization in LRRTM1/2-DKO mice compared with wild-type (WT) mice. Scale bar is 500 μm. (**B and D**) High-resolution confocal images revealed a reduction in punctate immunofluorescence for the excitatory presynapse marker VGlut1 in the CA1 dendritic layers oriens, radiatum and lacunosum, and increase in outer molecular layer of dentate gyrus (DG) in LRRTM1/2-DKO mice as compared with WT mice at 6 weeks postnatal. Scale bar is 10 μm for (**B**) and 20 μm for (**D**). (**C and E**) Quantitation of VGlut1 punctate integrated intensity per tissue area (Multiple t-test, *p<0.05, **p<0.01 comparing LRRTM1/2-DKO and wild-type mice for CA1 and DG, n=3–5 mice each after averaging data from 6 sections per mouse). Data presented as mean ± SEM.

The online version of this article includes the following source data and figure supplement(s) for figure 1:

**Source data 1.** Source data related to *Figure 1B, C*.

**Source data 2.** Source data related to *Figure 1D, E*.

**Figure supplement 1.** LRRTM1/2-DKO generation and confirmation of loss of LRRTM1 and LRRTM2.

**Figure supplement 1—source data 1.** *Figure 1—figure supplement 1* source blots.

**Figure supplement 2.** Levels of synaptic proteins in crude synaptosomal brain fractions of LRRTM1/2-DKO mice.

**Figure supplement 2—source data 1.** *Figure 1—figure supplement 2* source blots.

**Figure supplement 2—source data 2.** *Figure 1—figure supplement 2* source data.

**Figure supplement 3.** Inhibitory presynapse development is unaltered in CA1 and DG of LRRTM1/2-DKO.

**Figure supplement 3—source data 1.** *Figure 1—figure supplement 3* source data related to 1B.

**Figure supplement 3—source data 2.** *Figure 1—figure supplement 3* source data related to 1D.

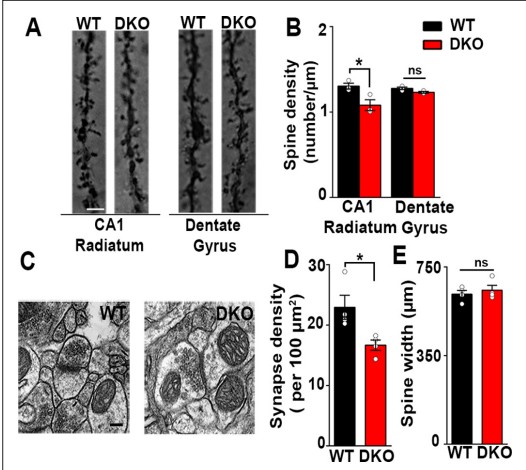

**Figure 2.** Excitatory synapses are reduced in the CA1 of LRRTM1/2-DKOs. (**A**) Golgi staining revealed a reduced density of dendritic spines in CA1 radiatum in the LRRTM1/2-DKO mice as compared with wild-type (WT) mice at 6 weeks postnatal. No differences were observed between genotypes in hippocampal dentate gyrus granule cell medial molecular layer. Scale bar is 1 μm. (**B**) Quantitation of dendritic spine density along CA1 radiatum and dentate gyrus medial molecular layer dendrites (Two-way ANOVA p<0.0001, *p<0.001, comparing LRRTM1/2-DKO and wild-type mice for each region by Bonferroni posthoc test, n=3 mice each for radiatum and n=3 mice each for outer molecular layer). (**C**) Representative electron micrographs from CA1 radiatum of wild-type (WT) and LRRTM1/2-DKO mice, scale bar 100 nm. (**D**) Quantitation of asymmetric synapse density (Student's t test *p=0.0394, n=4 mice each at 6 weeks postnatal). (**E**) Quantitation of spine width, Student's t test (p=0.3533, n=4 mice each at 6 weeks postnatal).

The online version of this article includes the following source data for figure 2:

**Source data 1.** Source data for *Figure 2A, B*.

**Source data 2.** Source data for *Figure 2D,E*.

LRRTM1 and LRRTM2 contribute to the development of excitatory presynapses differentially in hippocampal regions.

In the hippocampus, prominent populations of excitatory synapses are formed on dendritic spines of CA1 pyramidal neurons and on dentate gyrus granule cells (*Harris and Kater, 1994*; *Trommald and Hulleberg, 1997*). We measured the density of spines on CA1 pyramidal neurons in Golgi-stained brain sections, focusing on CA1 stratum radiatum, which predominantly contains inputs from CA3 Schaffer collaterals. Spine density was significantly reduced in CA1 pyramidal neurons of LRRTM1/2-DKOs as compared to WT cells (*Figure 2A and B*). In contrast, spine density in dentate gyrus granule cell medial molecular layer was unchanged in LRRTM1/2-DKOs (*Figure 2A and B*). Thus, LRRTM1 and LRRTM2 are required for the generation of normal numbers of dendritic spines in CA1 pyramidal neurons but are dispensable in this context in dentate gyrus granule cells.

Reductions in VGlut1 puncta immunofluorescence and spine density in the CA1 region of LRRTM1/2-DKOs indicate overall reductions in the number of excitatory synapses in this region. To assess synapse numbers by a more rigorous method, we used transmission electron microscopy. The density of asymmetric synapses, representing excitatory synapses, measured on dendritic spines was significantly reduced (27% decrease) in CA1 stratum radiatum of LRRTM1/2-DKOs as compared to WT controls (*Figure 2C and D*). As LRRTM1 and LRRTM2 are postsynaptic proteins, we also tested whether spine morphology was altered in LRRTM1/2-DKOs. However, spine width in the CA1 stratum radiatum was indistinguishable between LRRTM1/2-DKO and WT mice (*Figure 2E*). Overall, our data indicate that LRRTM1 and LRRTM2 are required for the normal development of excitatory synapses and spines in the CA1 region.

## Excitatory neurotransmission is differentially altered in the CA1 and dentate gyrus of LRRTM1/2-DKOs

We next tested whether the function of excitatory synapses is compromised in LRRTM1/2-DKOs. We performed whole-cell voltage clamp recordings from CA1 pyramidal neurons and dentate gyrus granule cells in hippocampal slices from LRRTM1/2-DKO and control WT mice. Miniature excitatory postsynaptic current (mEPSC) recordings from LRRTM1/2-DKO CA1 pyramidal neurons revealed a significant reduction in mEPSC frequency compared with control WT neurons (*Figure 3A and B*). mEPSC amplitudes were not significantly different between the groups (*Figure 3A and C*). In contrast, mEPSC frequency but not amplitude was increased in the LRRTM1/2-DKO dentate gyrus granule cells when compared to data from control WT mice (*Figure 3D–F*). Thus, LRRTM1 and LRRTM2 contribute to the development of functional excitatory synapses in CA1 pyramidal neurons but their loss in dentate gyrus during early development may be compensated by other synapse organizers. The changes in mEPSC frequency but not amplitude in the CA1 pyramidal neurons are consistent with

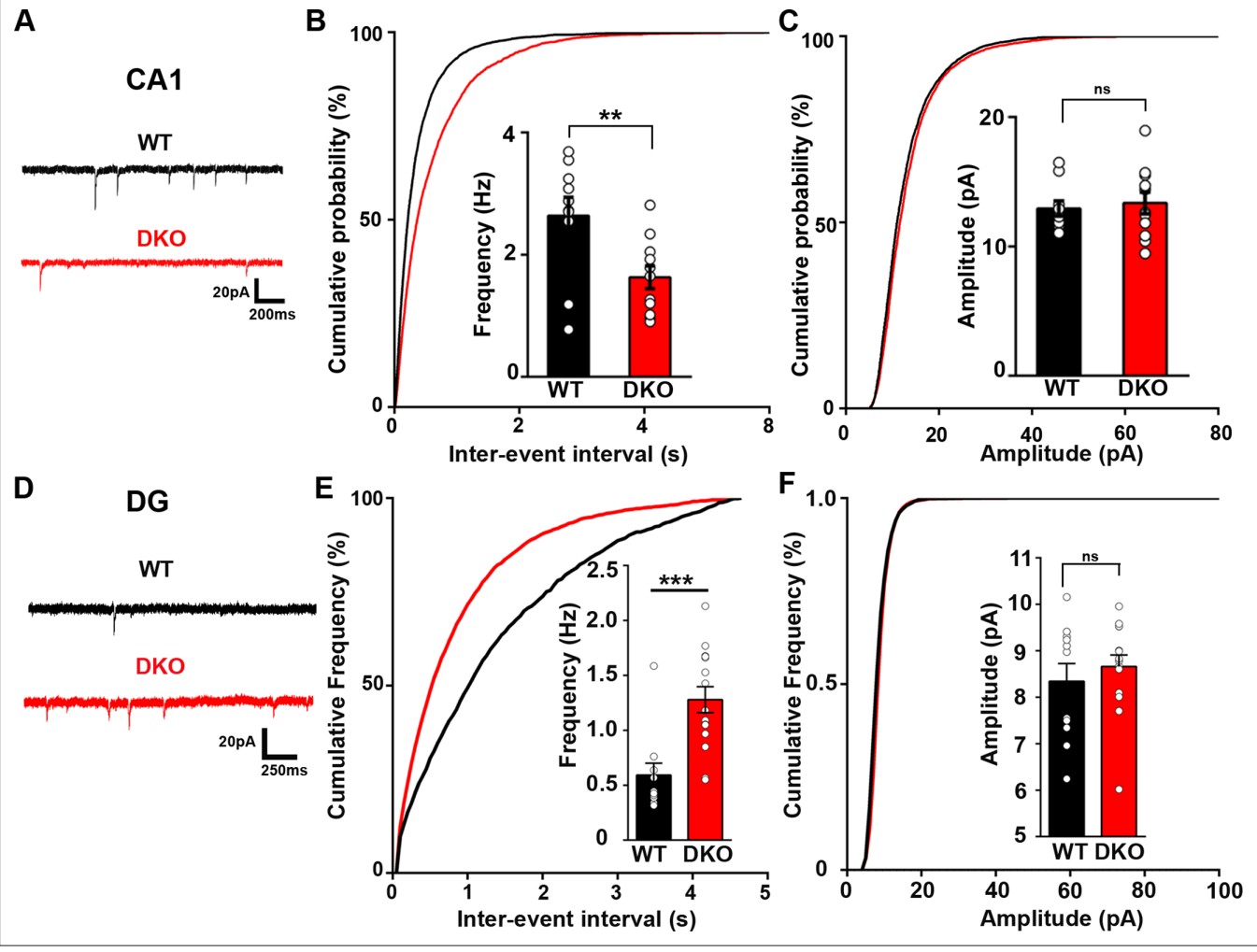

**Figure 3.** Excitatory synapse function is differentially altered in the CA1 and DG of LRRTM1/2-DKO mice. (**A and D**) Representative mEPSC recordings from wild-type (WT) and LRRTM1/2-DKO hippocampus CA1 pyramidal neurons and DG granule cells. (**B and C**) Cumulative distributions of mEPSC inter-event intervals (**B**) and amplitudes (**C**) in wild-type and LRRTM1/2-DKO neurons. Insets display mean ± SEM for mEPSC frequency (**B**) and amplitude (**C**) in CA1 pyramidal cells, respectively. Double knockout of *Lrrtm1* and *Lrrtm2* decreased mEPSC frequency, Student's t-test, \*\*p=0.0090, without affecting mEPSC amplitude, p=0.6873, n=10–11 neurons per group, three mice each. (**E and F**) Cumulative distributions of mEPSC inter-event intervals (**E**) and amplitudes (**F**) in wild-type and LRRTM1/2-DKO DG granule cells. Insets display mean ± SEM for mEPSC frequency (**E**) and amplitude (**F**) in DG granule cells, respectively. Double knockout of *Lrrtm1* and *Lrrtm2* increased mEPSC frequency, Student's t-test, \*\*\*p=0.0004, without affecting mEPSC amplitude, p=0.4731, n=11–15 neurons per group, three mice each.

The online version of this article includes the following source data for figure 3:

**Source data 1.** Source data for *Figure 3B, C*.

**Source data 2.** Source data for *Figure 3E, F*.

corresponding changes in imaging and electron microscopy data, which indicated roles for LRRTM1 and LRRTM2 in controlling the number of excitatory synapses in CA1 pyramidal neurons.

## LRRTM1 and LRRTM2 regulate hippocampal synaptic plasticity and memory

We assessed presynaptic function in CA1 pyramidal neurons of LRRTM1/2-DKOs. We observed modest changes in paired-pulse facilitation at more delayed inter-stimulus intervals (ISIs) in LRRTM1/2-DKO compared to WT controls (*Figure 4—figure supplement 1*). Next, we assessed LTP at CA3-CA1 synapses after chronic deletion of LRRTM1 and LRRTM2. LTP was induced with a single train of high frequency stimulation (HFS; 1 s at 100 Hz). Both WT and LRRTM1/2-DKO slices exhibited an

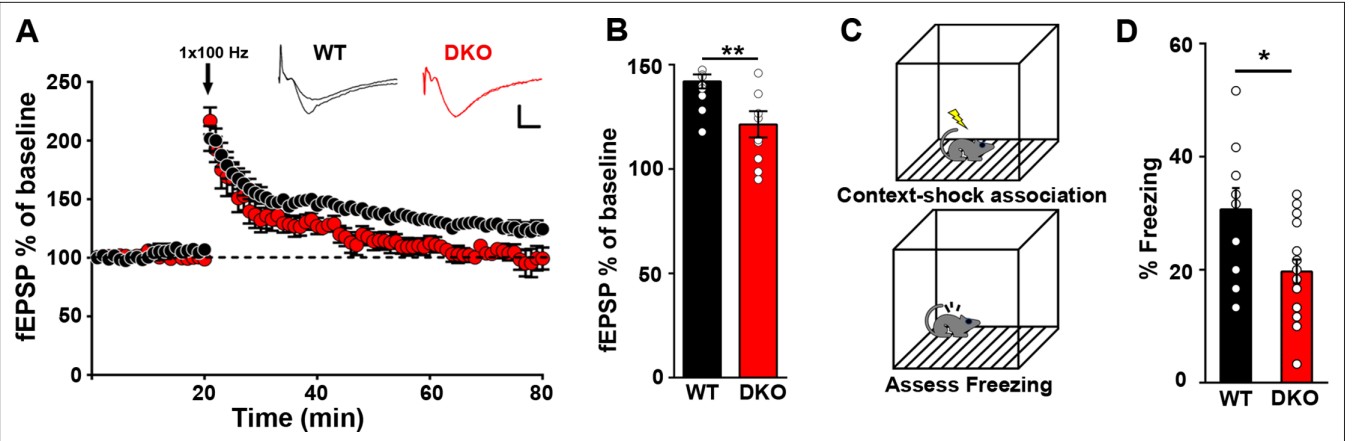

**Figure 4.** Impaired hippocampal LTP with hippocampal dependent memory in LRRTM1/2-DKO mice. (**A**) The maintenance of long-term potentiation (LTP) is reduced in LRRTM1/2-DKO mouse CA1. Inset: representative fEPSP recordings from wild-type and LRRTM1/2-DKO stratum radiatum CA3-CA1 synapses in acute hippocampal slices taken 10 min into baseline and 55 min after high-frequency stimulation (HFS; 1 sec at 100 Hz). (**B**) Comparisons of average fEPSP% values LRRTM1/2-DKO and WT mice, (Student's t-test **p=0.0071, n=10–12 slices per group, three mice each). (**C**) Diagrammatic representation of contextual fear conditioning protocol where freezing was recorded in both LRRTM1/2-DKO and control groups of mice 24 hr after receiving single (0.7 mA) shocks paired with the contextual conditioning chamber. (**D**) Contextual memory is impaired in LRRTM1/2-DKO mice. Percentage of time spent freezing following contextual fear conditioning (CFC) training was compared between WT and LRRTM1/2-DKO mice. LRRTM1/2-DKO showed significantly reduced levels of freezing when exposed to the CFC chamber 24 hr after initial training (Student's t-test, *p=0.0189, n=10 mice for WT and n=16 mice for LRRTM1/2-DKO).

The online version of this article includes the following source data and figure supplement(s) for figure 4:

**Source data 1.** Source data for *Figure 4A, B*.

**Source data 2.** *Figure 4D* source data.

**Figure supplement 1.** Presynaptic function in CA1 of LRRTM1/2-DKO mice.

**Figure supplement 1—source data 1.** *Figure 4—figure supplement 1* source data.

initial potentiation of the average field excitatory postsynaptic potential (fEPSP) (*Figure 4A*). However, whereas fEPSP values recorded 40–60 min after HFS revealed potentiation to 126% ± 4% of baseline in slices from WT mice, a significant loss of potentiation, with fEPSP values at 106% ± 6% of baseline, was observed in LRRTM1/2-DKO slices (*Figure 4A and B*). This deficit at later but not early phases following LTP induction is consistent with a role of LRRTM1 and LRRTM2 in the postsynaptic maintenance of synaptic potentiation. These results also indicate that the LTP deficits caused by the loss of LRRTM1 and LRRTM2 in CA1 cannot be compensated by other proteins.

Impaired hippocampal LTP has been linked to reduced performance in behavioral memory tasks (*Nabavi et al., 2014*; *Orsini and Maren, 2012*; *Schafe et al., 2001*). Moreover, reduced synapse numbers in the CA1 and altered synaptic transmission in the CA1 and dentate gyrus of LRRTM1/2-DKOs likely influence hippocampal network function. To determine if hippocampus-dependent memory is compromised in LRRTM1/2-DKOs, we used a classical fear conditioning paradigm to assess contextual memory (*Figure 4C*). Mice were trained to associate a foot shock with a particular context, and learning and memory was assessed as extent of freezing in response to subsequent exposure to the context only. At 24 hr post-training, LRRTM1/2-DKOs exhibited significantly reduced levels of freezing as compared to WT controls (*Figure 4D*). These data indicate that LRRTM1 and LRRTM2 are required for the maintenance of long-term contextual memories.

## Local deletion of LRRTM1/2 in the CA1 of developing brain impairs excitatory neurotransmission

Although excitatory synaptic transmission in the CA1 of mice lacking LRRTM1 and LRRTM2 globally (LRRTM1/2-DKO) is impaired, the global deletion approach does not allow one to distinguish whether the observed deficits are due to presynaptic or postsynaptic changes. To resolve this issue, we deleted *Lrrtm1* and *Lrrtm2* selectively in the CA1 of postnatal 0 (P0) mice. To do so, we delivered AAV8-hSyn-Cre-eGFP (P0-LRRTM1/2-cDKO) or AAV8-hSyn-eGFP (control) to the dorsal CA1 of P0 *Lrrtm1^floxed/*

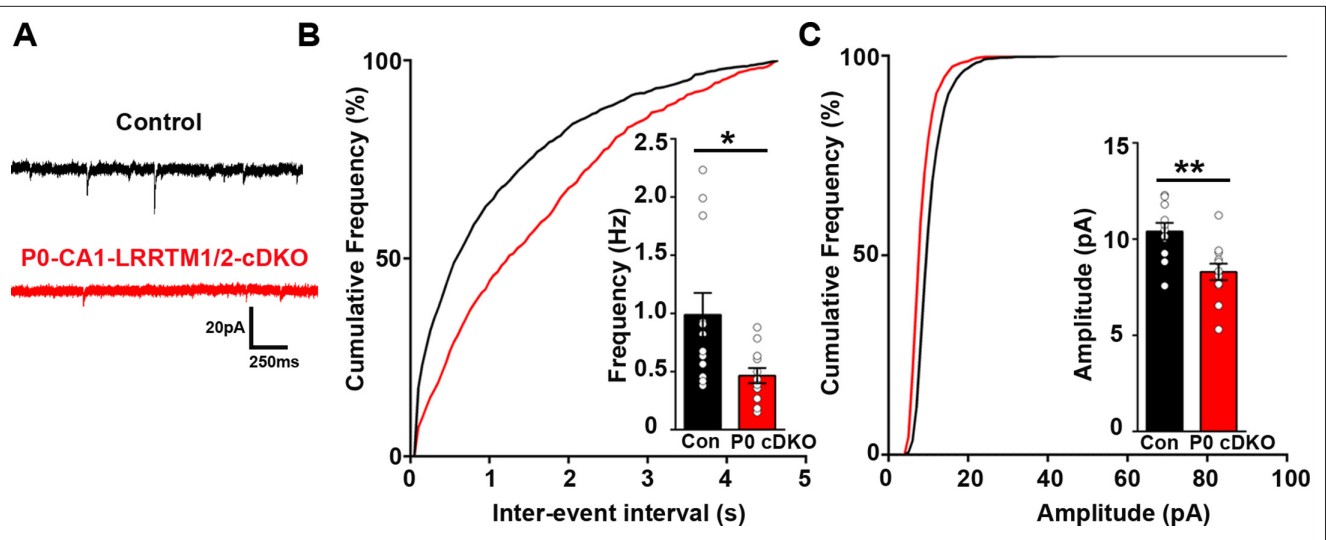

**Figure 5.** Excitatory synapse function is impaired in the CA1 of P0-CA1-LRRTM1/2-cDKO mice. (**A**) Representative mEPSC recordings from control and P0 LRRTM1/2-cDKO hippocampal CA1 pyramidal neurons. (**B and C**) Cumulative distributions of mEPSC inter-event intervals (**B**) and amplitudes (**C**) in control and P0-CA1-LRRTM1/2-cDKO neurons. Insets display mean ± SEM for mEPSC frequency (**B**) and amplitude (**C**) in CA1 pyramidal cells, respectively. Double knockout of *Lrrtm1* and *Lrrtm2* at P0 decreased mEPSC frequency and amplitude (Student's t-test, *p=0.0157 for frequency and **p=0.0026 for amplitude, n=12–13 per group, three mice each).

The online version of this article includes the following source data and figure supplement(s) for figure 5:

**Source data 1.** *Figure 5* source data for 5B and C.

**Figure supplement 1.** Region-specific injection in CA1 of LRRTM1/2$^{floxed/floxed}$ mice in P0 pups to obtain P0-CA1-LRRTM1/2-cDKO mice.

---

$^{floxed}$/*Lrrtm2$^{floxed/floxed}$* mice (***Figure 5—figure supplement 1***). All analyses were done at 6–7 weeks of age, the same age as for analyses of the global LRRTM1/2-DKO. We performed whole-cell voltage clamp recordings from CA1 pyramidal neurons of P0-LRRTM1/2-cDKO and control mice. Similar to the LRRTM1/2-DKO mice, mEPSC frequency was significantly reduced in the CA1 pyramidal neurons of P0-LRRTM1/2-cDKO as compared to control mice (***Figure 5A and B***). A modest reduction in mEPSC amplitude was also observed in the P0-LRRTM1/2-cDKOs (***Figure 5C***). Thus, reduced synaptic transmission in the CA1 of P0-LRRTM1/2-cDKO are due to postsynaptic deficits.

## LRRTM1 and LRRTM2 are not required for maintaining synapse numbers in the mature brain

Next, we investigated the role of LRRTM1 and LRRTM2 in synapse development and plasticity in the mature brain. We therefore generated mice in which *Lrrtm1* and *Lrrtm2* were acutely deleted in the adult hippocampus. To do so, we delivered AAV8-hSyn-Cre-eGFP (LRRTM1/2-cDKO) or AAV8-hSyn-eGFP (control) to the dorsal hippocampus of 3 weeks old *Lrrtm1$^{floxed/floxed}$*/*Lrrtm2$^{floxed/floxed}$* mice (***Figure 6A and B***). Given that the half-lifes of LRRTM1 and LRRTM2 are approximately 3 and 7 days, respectively, in mouse cortical synapses (***Fornasiero et al., 2018***), all analyses were done 3 weeks after virus delivery which should allow for the effective expression of Cre recombinase and deletion of the targeted genes. Thus, all analyses were done at 6–7 weeks of age, the same age as for the Global DKO and Local early cDKO. We obtained ~98% (dorsal CA1) and ~90% (dorsal dentate gyrus) maximal infection efficiency at the site of injection (average efficiency of infection in dorsal hippocampus was ~75%), as assessed by GFP$^{+ve}$ DAPI cells (***Figure 6C***). Both LRRTM1 and LRRTM2 were markedly depleted in the CA1 of LRRTM1/2-cDKO when compared to that in controls (***Figure 6—figure supplement 1***). All analyses for synapse numbers and plasticity were done near the sites of maximal infection. Virus spread into the ventral CA1 and dentate gyrus was significantly lower (~10%), thus allowing us to selectively assess the functions of LRRTM1 and LRRTM2 in the mature dorsal CA1 and dentate gyrus.

We first performed apotome-enabled optical sectioning and imaging of VGlut1 and GAD65 in CA1 and DG dendritic layers. Quantitative analysis revealed that puncta immunofluorescence for VGlut1

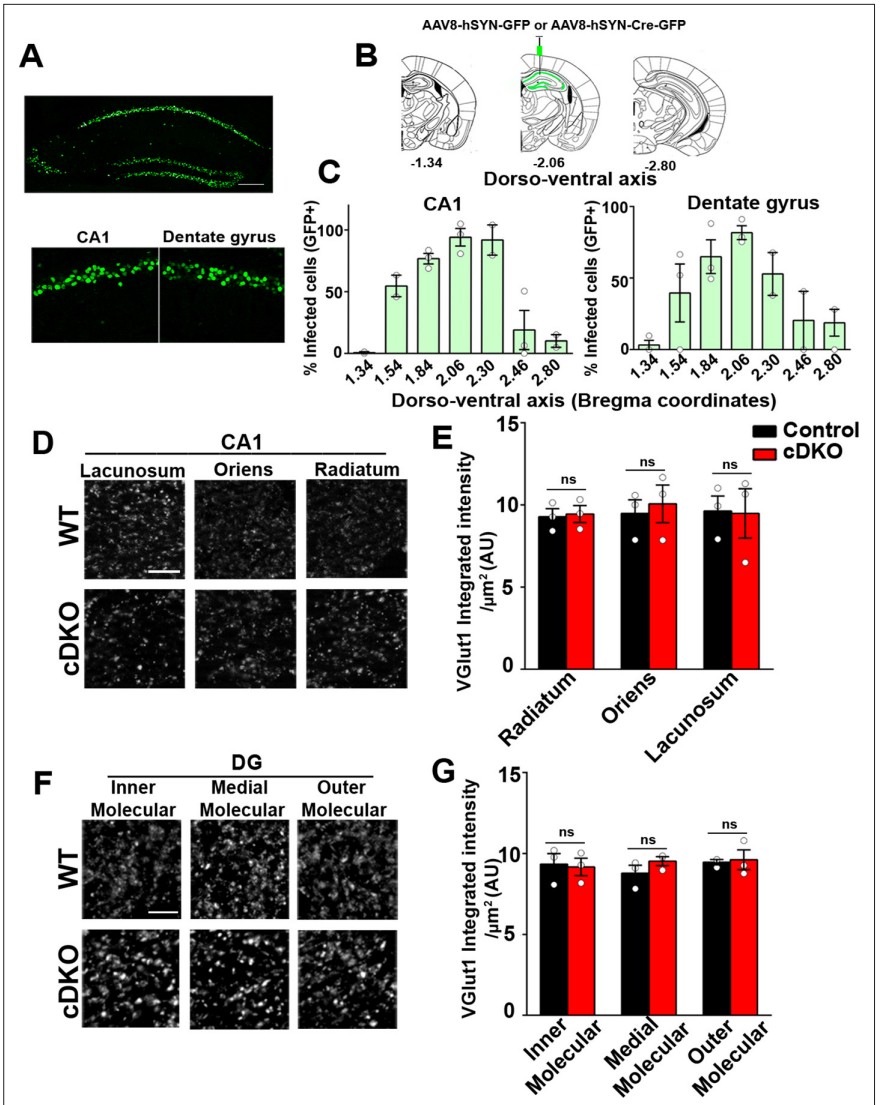

**Figure 6.** Excitatory presynapse development is unaltered in CA1 dendritic layers and DG molecular layers of LRRTM1/2-cDKO. (**A**) Representative image of dorsal hippocampus (injection site) with infected CA1 and dentate gyrus neurons (green) from AAV8-hSYN-Cre-eGFP. Below, zoomed in images of the dorsal CA1 and dentate gyrus. Scale bar is 20 μm. (**B**) Representative images of whole brain (modified from Allen Brain Institute) showing bregma points −1.34, −2.06, and −2.80, with 2.06 AP as the injection site in LRRTM1/2-cDKO mice and controls. (**C**) Percent infected cells (Number of GFP +neurons / Number of DAPI +neurons) across the dorso-ventral axis for CA1 and dentate gyrus. (**D and F**) High-resolution immunofluorescence of VGlut1 punctate was unaltered in the dendritic layers of CA1 and DG molecular layers in LRRTM1/2-cDKO mice as compared with control mice. Scale bar is 5 μm. (**E and G**) Quantitation of VGlut1 punctate integrated intensity per tissue area, (Multiple t-test, n=3 mice each after averaging data from 6 sections per mouse). Data presented as mean ± SEM.

The online version of this article includes the following source data and figure supplement(s) for figure 6:

**Source data 1.** *Figure 6* source data for 6C.

**Source data 2.** *Figure 6* source data for 6DG.

**Figure supplement 1.** LRRTM1 and LRRTM2 knockout confirmation.

**Figure supplement 2.** Inhibitory presynapse development is unaltered in CA1 and DG of LRRTM1/2-cDKO mice.

**Figure supplement 2—source data 1.** *Figure 6—figure supplement 2* source data.

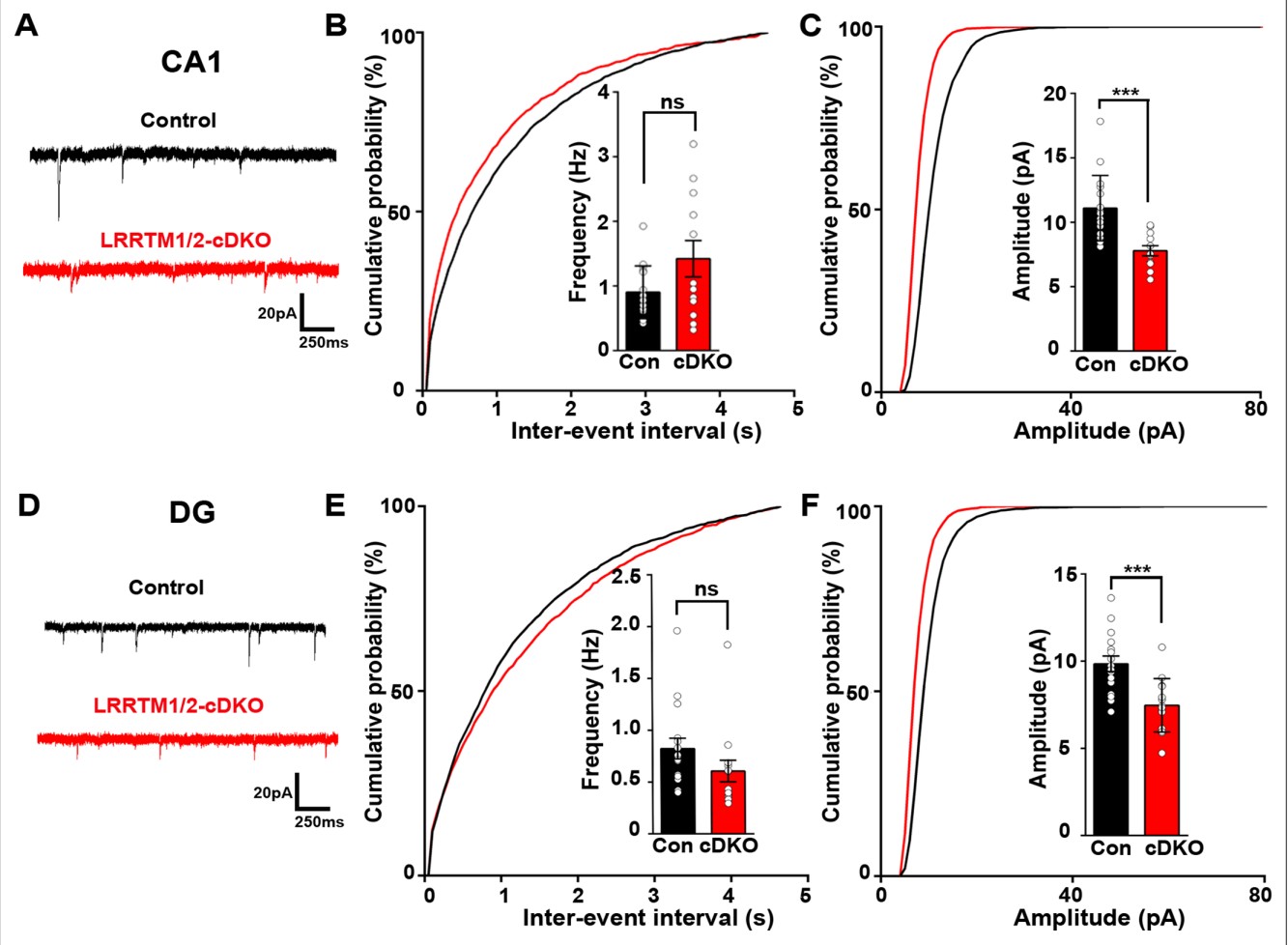

**Figure 7.** Excitatory synapse function is impaired in the CA1 and DG of LRRTM1/2-cDKO mice. (**A and D**) Representative mEPSC recordings from control and LRRTM1/2-DKO hippocampus CA1 pyramidal neurons and DG granule cells. (**B and C**) Cumulative distributions of mEPSC inter-event intervals (**B**) and amplitudes (**C**) in control and LRRTM1/2-DKO CA1 pyramidal neurons. Insets display mean ± SEM for mEPSC frequency (**B**) and amplitude (**C**) in CA1 pyramidal cells, respectively. Acute conditional deletion of *Lrrtm1* and *Lrrtm2* in CA1 decreased mEPSC amplitude (Student's t-test, ***p=0.0004, without affecting mEPSC frequency, p=0.0651. n=16 neurons for three control mice and n=12 neurons for three LRRTM1/2-cDKO mice). (**E and F**) Cumulative distributions of mEPSC inter-event intervals (**E**) and amplitudes (**F**) in control and LRRTM1/2-cDKO neurons. Insets display mean ± SEM for mEPSC frequency (**E**) and amplitude (**F**) in DG granule cells, respectively. Acute conditional deletion of *Lrrtm1* and *Lrrtm2* in DG decreased mEPSC amplitude (Student's t-test, ***p=0.0006, without affecting mEPSC frequency, p=0.1444, n=16 neurons for three control mice and n=14 neurons for three LRRTM1/2-cDKO mice).

The online version of this article includes the following source data for figure 7:

**Source data 1.** *Figure 7* source data.

was comparable between the LRRTM1/2-cDKO and control mice in all CA1 (stratum oriens, stratum radiatum, stratum lacunosum moleculare) and dentate gyrus layers (inner, medial, outer molecular layer) (*Figure 6D–G*). Likewise, GAD65 puncta immunofluorescence was comparable between LRRTM1/2-cDKO and control mice in all CA1 and DG dendritic layers (*Figure 6—figure supplement 2*). Thus, while required for synapse formation in the developing brain, LRRTM1 and LRRTM2 are dispensable for maintaining synapse numbers in the mature brain.

Next, we investigated the role of LRRTM1 and LRRTM2 in synaptic transmission in the mature CA1 and dentate gyrus. Consistent with our previous report (*Bhouri et al., 2018*), whereas LRRTM1 and LRRTM2 are required for normal mEPSC frequency in the CA1 of developing brain without affecting mEPSC amplitudes (*Figure 3A–C*), the opposite is the case in the mature circuit, where LRRTM1 and LRRTM2 loss in both CA1 and DG affects the amplitude but not the frequency of mEPSCs

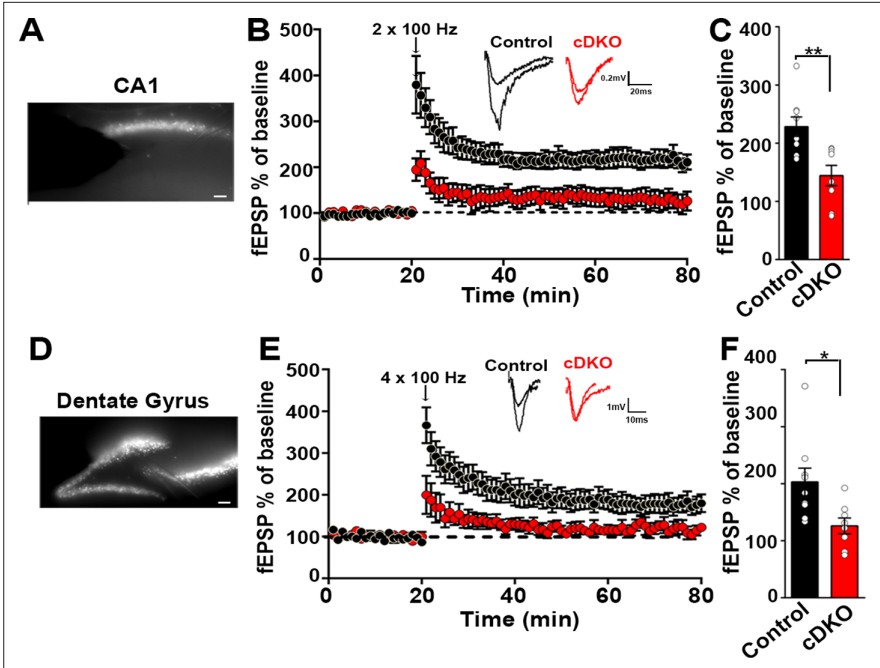

**Figure 8.** Impaired plasticity in dorsal hippocampal CA1 and dentate gyrus regions in LRRTM1/2-cDKO mice. (**A**) Epifluoroscent image of dorsal CA1 with stimulating and recording electrodes. Scale bar 50 μm. (**B**) fEPSP recordings from control and LRRTM1/2-cDKO stratum radiatum CA3-CA1 synapses in acute hippocampal slices taken 20 min into baseline and 60 min after high-frequency stimulation (2 × 1 s at 100 Hz). Insets: Representative traces. (**C**) Comparisons of %fEPSP values taken from significant reductions in the fEPSPs in CA1 region of LRRTM1/2-cDKO mice, (Student's t-test **p=0.0033, n=9 slices from three control mice and n=8 slices from three LRRTM1/2-cDKO mice). (**D**) Epifluoroscent image of dorsal dentate gyrus with stimulating and recording electrodes. Scale bar 50 μm. (**E**) fEPSP recordings from control and LRRTM1/2-cDKO inner molecular layer of dentate gyrus synapses in acute hippocampal slices taken 20 min into baseline and 60 min after high-frequency stimulation (4 × 1 s at 100 Hz). Insets: Representative traces. (**F**) Comparisons of average %fEPSP values revealed significant reductions in fEPSPs in dentate gyrus of LRRTM1/2-cKO mice, (Student's t test *p=0.0176, n=9 slices for three control mice and n=8 slices for three LRRTM1/2-cDKO mice). Data presented as mean ± SEM.

The online version of this article includes the following source data and figure supplement(s) for figure 8:

**Source data 1.** *Figure 8* source data.

**Figure supplement 1.** Presynaptic function is unaltered in CA1 of LRRTM1/2-cDKO mice.

**Figure supplement 1—source data 1.** *Figure 8—figure supplement 1* source data.

---

(*Figure 7A–F*). Thus, LRRTM1 and LRRTM2 regulate synaptic transmission differentially in the developing and the mature brain.

## LRRTM1 and LRRTM2 are required for LTP in multiple neural circuits in the mature brain

Knockout of *Lrrtm1* and *Lrrtm2* in sparsely targeted adult hippocampal CA1 impaired LTP in individual pyramidal neurons (*Bhouri et al., 2018*). As we achieved an average of ~85% virus spread across the dorsal CA1 (dorsoventral axis –1.54 to –2.30 from bregma), we investigated LTP upon widespread KO of LRRTM1 and LRRTM2. We assessed LTP using field recordings of evoked fEPSPs. LTP was induced by high frequency stimulation of Schaffer collaterals (2 × 100 Hz, 20 s interval) and fEPSPs were recorded. Whereas fEPSPs after stimulation increased to 228% ± 16.57% in control mice, LTP was significantly impaired in LRRTM1/2-cDKO mice (144.4% ± 17.5%) (*Figure 8A–C*). To assess whether pre-synaptic function may be compromised in the CA1 of LRRTM1/2-cDKO mice, we examined paired-pulse facilitation (*Figure 8—figure supplement 1*). There was no difference between control and LRRTM1/2-cDKO mice suggesting that mechanisms underlying short-term facilitation are not impacted by local deletion of *Lrrtm1* and *Lrrtm2* in the mature CA1.

Next, we assessed whether acute deletion of *Lrrtm1* and *Lrrtm2* alters LTP in the dentate gyrus, a region with strong co-expression of LRRTM3 and LRRTM4. We achieved an average of ~65% virus spread across the dorsal dentate gyrus (dorsoventral axis –1.54 to –2.30 from bregma). We induced LTP in the inner molecular layer of the dentate gyrus with a high frequency stimulation (4 × 100 Hz, 5 min interval). Whereas fEPSPs after stimulation increased to 203.1% ± 24% in control mice, LTP was significantly impaired in LRRTM1/2-cDKO mice (126% ± 13.84%) (*Figure 8D–F*). These results show that LRRTM3 and LRRTM4 cannot functionally compensate the effect of the loss of LRRTM1 and LRRTM2 on LTP, indicating that LRRTM1 and LRRTM2 control LTP by a distinct mechanism.

## LRRTM1 and LRRTM2 are required for dorsal hippocampus-dependent cognitive function in the mature brain

The effective loss of LRRTM1 and LRRTM2 in vivo significantly reduced both basal synaptic strength and LTP in the dorsal CA1 region and the DG. These deficits could alter network function and consequently alter cognitive functions associated with the dorsal hippocampus (*Fanselow and Dong, 2010*).

A prerequisite for assessing cognitive function is normal locomotor activity and motor coordination. To assess gross motor skills, we used an accelerating rotarod with an acceleration rate of 20 rpm/min. The latency to fall was comparable between the LRRTM1/2-cDKO and control groups (*Figure 9—figure supplement 1A*). We then assessed locomotor performance using the open field test. Average speed and distance travelled were again comparable between LRRTM1/2-cDKOs and control animals (*Figure 9—figure supplement 1C and D*), indicating that locomotion is not affected by acute deletion of *Lrrtm1* and *Lrrtm2* in the dorsal hippocampus. The open-field test is frequently used to infer risk-associated anxiety levels in mice (*Seibenhener and Wooten, 2015*) as mice with high anxiety levels spend considerably less time in the inner zone of the open field. Time spent in the inner and outer zones of the open field was equivalent between the genotyopes (*Figure 9—figure supplement 1B*), indicating that deletion of *Lrrtm1* and *Lrrtm2* in the dorsal hippocampus does not increase risk-associated anxiety behavior. We then used the more sensitive elevated plus maze (EPM) test to assess avoidance-related anxiety behavior, where mice are allowed to explore open and closed arms of the EPM. The EPM test utilizes the innate behavior of mice to prefer enclosed spaces and their tendency to avoid heights and open areas; the open arms of the EPM are more anxiogenic than the closed arms (*Walf and Frye, 2007*). The time spent in and the number of entries into the open arms were comparable between LRRTM1/2-cDKO and control mice (*Figure 9—figure supplement 1E and F*). The total distance travelled and the average speed in the EPM were also comparable between the two groups (*Figure 9—figure supplement 1G and H*). Thus, conditional deletion of *Lrrtm1* and *Lrrtm2* in dorsal hippocampus of adult mice does not alter avoidance-related anxiety behavior in mice.

Next, we assessed social memory in the LRRTM1/2-cDKO and control groups using Crawley's three chambered sociability test (*Kaidanovich-Beilin et al., 2011*; *Figure 9A*). Social memory is primarily associated with the ventral hippocampus along with amygdala and hypothalamus (*Okuyama et al., 2016*). In Crawley's sociability and preference test, mice can freely choose to interact with familiar and nonfamiliar conspecifics. The first block of the test included 10 min habituation for the test mice in all three chambers of the test. In the second block, Stranger I was introduced in one of the chambers and interaction of control and LRRTM1/2-cDKO mice with Stranger I was assessed. Both groups spent more time interacting with Stranger I as compared to the middle and empty chambers (*Figure 9B*). In the third block, a second mouse (Stranger II) was introduced in the third chamber. Both control and LRRTM1/2-cDKO mice interacted more with Stranger II as compared to Stranger I (*Figure 9C*). Thus, Lrrtm1/2-cDKO do not differ from control mice in sociability or social novelty preference behavior.

Context acquisition through the hippocampus requires integration of various visuo-spatial, auditory, and olfactory cues (*Fanselow, 2000*; *Sanders et al., 2003*; *Wiltgen et al., 2006*). These cues are acquired by exploratory behavior encoded in cortical structures, such as subiculum, and nucleus accumbens which send projections to the dorsal hippocampus and are consolidated in the prefrontal cortex (*Fanselow, 2000*). In both, LRRTM1/2-DKO and -cDKO mice, LTP is impaired in the hippocampus (*Figures 4A and 8B*). In hippocampus, LTP maintenance is required for coding integral representation of contexts and its association with unconditional stimuli (*Nicoll, 2017*; *Penn et al., 2017*; *Takeuchi et al., 2014*). Therefore, we examined hippocampus-dependent contextual fear conditioning to assess long term memory encoding in LRRTM1/2-cDKO and control mice. Mice were pre-exposed to a context 24 hr before a mild foot shock, and were then trained to associate mild foot shocks with a

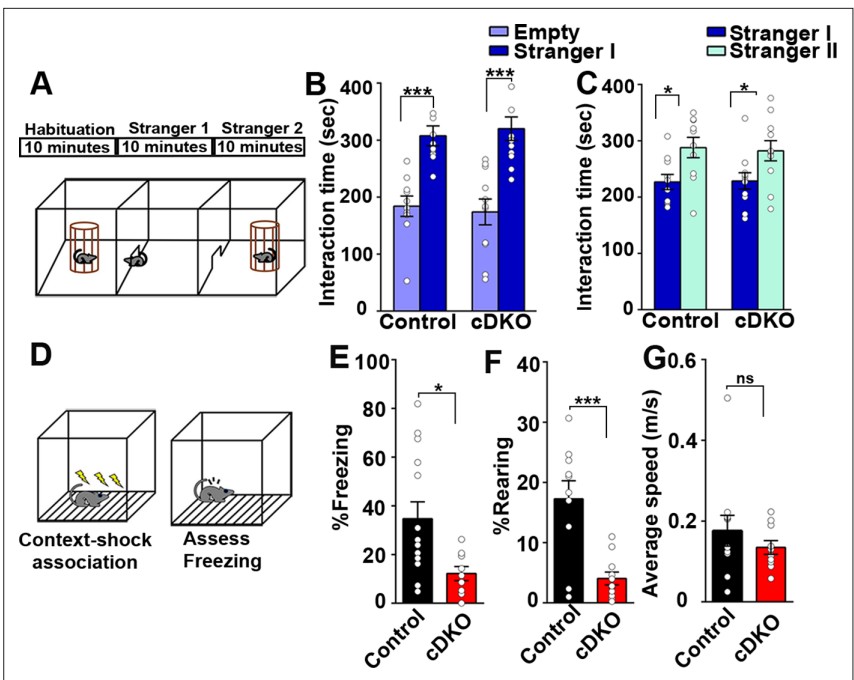

**Figure 9.** Contextual memory is impaired in LRRTM1/2-cDKO mice. (**A**) Experimental scheme of three-chambered social approach. (**B and C**) Three chambered social approach test was used to test social memory between controls and LRRTM1/2-cDKO mice. (**B**) Social novelty (time spent with introduced stranger mice versus empty chamber) was not altered between controls and LRRTM1/2-cDKOs, (Student's t-test for control, ***p<0.0001 and LRRTM1/2-cDKO, ***p<0.0001, n=11 mice for each group). (**C**) Interaction time spent with familiar mice and novel mice was comparable between control and LRRTM1/2-cDKO as they spent more time interacting with novel mice (Student's t-test for control, *p=0.0144 and LRRTM1/2-cDKO, *p=0.0307, n=11 mice for each group). (**D**) Experimental scheme of contextual fear memory test. Mice were pre-exposed to context on day 1 followed by aversive shocks on day 2. Freezing and rearing behavior was assessed on day 3. (**E–G**) Percentage of time spent freezing and rearing following contextual fear conditioning (CFC) training was compared between control and LRRTM1/2-cDKO mice 24 hr after receiving three trains of 1.0 mA shocks. LRRTM1/2-cDKO showed significantly reduced levels of freezing and rearing when exposed to the CFC chamber 24 hr after initial training (n=14 mice for control, n=11 mice for LRRTM1/2-cDKO), (Student's t-test, *p=0.0212 and ***p=0.0004). (**F**) Average speed was comparable between the groups, (Student's t-test, p=0.3500). Data presented as mean ± SEM.

The online version of this article includes the following source data and figure supplement(s) for figure 9:

**Source data 1.** *Figure 9* source data.

**Figure supplement 1.** Behavioral tests performed on LRRTM1/2-cDKO mice showed no differences in motor skills and anxiety-like behavior.

**Figure supplement 1—source data 1.** *Figure 9—figure supplement 1* source data.

particular context. Learning and memory were assessed as percent freezing and rearing in response to subsequent exposure to the context alone (*Figure 9D*). LRRTM1/2-cDKO mice exhibited significantly reduced levels of freezing and rearing as compared to controls (*Figure 9E and F*), but their average speed was comparable to control values (*Figure 9G*). Thus, conditional deletion of *Lrrtm1* and *Lrrtm2* in dorsal hippocampus causes deficits in hippocampus-dependent long-term contextual memory.

## Discussion

We demonstrate here, using global and region-restricted genetic approaches combined with multiple independent analyses, that LRRTM1 and LRRTM2 have differential and context-dependent functions in the organization and plasticity of synapses and in cognition. For comprehensive analysis, we generated and analyzed three models: (1) Global early deletion model (LRRTM1/2-DKO); (2) Local early deletion (P0-LRRTM1/2-cDKO) and; (3) Local late deletion (LRRTM1/2-cDKO).

Our investigations led us to the following conclusions. First, LRRTM1 and LRRTM2 are important for the normal development of excitatory synapses, selectively in the hippocampal CA1 region. Second, inhibitory synapses throughout the hippocampus and are not affected by the loss of LRRTM1 and LRRTM2. Third, LRRTM1 and LRRTM2 are dispensable for the maintenance of normal numbers of excitatory synapses in mature neurons. Fourth, synaptic deficits due to LRRTM1 and LRRTM2 deletion are postsynaptic in origin. Fifth, LRRTM1 and LRRTM2 are required for LTP at CA3-CA1 synapses in both developing and mature neural circuits. Surprisingly, in this context, LRRTM1 and LRRTM2 are also required for LTP at dentate gyrus synapses. These results indicate that the roles of LRRTM1 and LRRTM2 in synapse development and plasticity are functionally dissociable. Finally, our results show that LRRTM1 and LRRTM2 are essential for enduring hippocampus-dependent long-term contextual memory.

There is substantial discord in the LRRTM literature due to conflicting data obtained in LRRTM KD experiments (*de Wit et al., 2009*; *Ko et al., 2011*; *Soler-Llavina et al., 2011*). Whereas in one study, shRNA-mediated KD of LRRTM2 in hippocampal neuron cultures was shown to significantly reduce synapse numbers (*de Wit et al., 2009*), a different study indicated that shRNA-mediated KD of LRRTM1 and LRRTM2 in hippocampal cultures does not alter synapse numbers (*Ko et al., 2011*). When combined with neuroligin-3 KD in neuroligin-1 KO neurons, however, shRNA-mediated KD of LRRTM1 and LRRTM2 caused reductions in the number of excitatory synapses in hippocampal cultures but did not reduce spine density in CA1 neurons in vivo (*Soler-Llavina et al., 2011*). Our findings provide clarity and resolve some of these discrepancies. In contrast to previous studies, we found that constitutive deletion of *Lrrtm1* and *Lrrtm2* causes significant reductions in the numbers of asymmetric synapses, VGlut1 puncta fluorescence, spine density, and mEPSC frequency in CA1 neurons in vivo. Likely reasons for these discrepancies include differences between in vitro and in vivo settings and problems with the specificity, efficacy, and timing of KD approaches. As opposed to our constitutive deletion approach, acute deletion of *Lrrtm1* and *Lrrtm2* in mature neurons did not alter the number of synapses but led to significantly reduced mEPSC amplitudes (*Figure 7*; *Bhouri et al., 2018*). However, consistent with the constitutive deletion approach, local early deletion of *Lrrtm1* and *Lrrtm2* in the CA1 (P0-LRRTM1/2-cDKO) significantly reduced mEPSC frequency, and was accompanied with modest decrease in mEPSC amplitude (*Figure 5*). These results indicate that LRRTM1 and LRRTM2 mediate distinct functions in developing and mature neurons. Specifically, during development, LRRTM1 and LRRTM2 contribute to synapse development whereas in the mature brain, consistent with previous reports (*Bhouri et al., 2018*; *Karimi et al., 2021*; *Ramsey et al., 2021*), they contribute to maintaining the complement of AMPARs at synapses.

A key open question in our understanding of neural circuit formation concerns the molecular factors that determine the regional, or cell-type specificity of synapse formation. We show that the constitutive loss of *Lrrtm1* and *Lrrtm2* leads to reduced synapse numbers selectively in the CA1 region. Our results are in agreement with previous studies showing dispersed synaptic vesicles or slightly reduced synapse numbers in specific laminae of CA1 in mice lacking *Lrrtm1* (*Linhoff et al., 2009*; *Schroeder et al., 2018*; *Takashima et al., 2011*). Loss of *Lrrtm3* or *Lrrtm4*, or a knock-in mutation in *Lrrtm4* that blocks its interaction with heparan sulfate, selectively reduce synapse numbers in the dentate gyrus but not in the CA1 region (*de Wit et al., 2013*; *Roppongi et al., 2020*; *Siddiqui et al., 2013*; *Um et al., 2016*). In contrast, deletions of several other organizers of excitatory synapses have either no or much more widespread effects on synapse numbers. Synapse numbers in mice lacking neuroligin 1 or neuroligin 3 are not altered (*Blundell et al., 2010*; *Varoqueaux et al., 2006*), and even in cultured neurons and the brain of newborn neuroligin 1–3 triple KO mice, synapse numbers are unchanged (*Varoqueaux et al., 2006*). Hence, neuroligins are thought to function in synapse maturation and plasticity but not in initial synapse formation (*Südhof, 2017*). On the other hand, mice lacking IL1RacP, for instance, have reduced numbers of excitatory synapses and reduced spine density in all brain regions analyzed (*Yoshida et al., 2012*), and EphB1-3 triple KO mice exhibit widespread reductions in the numbers of excitatory synapse numbers and dendritic spines, for example in cortex and hippocampus (*Henkemeyer et al., 2003*; *Kayser et al., 2006*). These more widespread functions of IL1RacP and EphB1-3 contrast with the cell selective functions of LRRTMs.

LRRTMs are co-expressed with multiple other synapse organizers in both pyramidal neurons and granule cells. In the developing hippocampus (Global DKO), we observed a 25–30% loss of excitatory synapses in the CA1 indicating that in pyramidal neurons, LRRTM1/2 are important drivers of

synapse development. However, CA1 pyramidal neurons express multiple other postsynaptic organizers, including neuroligins, which likely account for significant functional compensation for the loss of LRRTMs in these cell types.

LRRTMs have discrete expression profiles, and levels of individual LRRTMs in different brain regions and cell types vary substantially. In the hippocampus, LRRTM1 and LRRTM2 are the only LRRTM family members expressed in CA1, whereas all LRRTMs are expressed in the dentate gyrus, with LRRTM4 expression exhibiting a particularly high level (*Laurén et al., 2003*). Given their sequence homology and common associations with AMPA receptors and PSD-95 family scaffold proteins (*de Wit et al., 2009*; *Linhoff et al., 2009*; *Siddiqui et al., 2013*; *Soler-Llavina et al., 2013*), it is likely that in developing dentate gyrus neuronal circuits, LRRTM3 and LRRTM4 can compensate for the loss of LRRTM1 and LRRTM2. Indeed, loss of LRRTM3 or LRRTM4 substantially reduces synapse numbers in the dentate gyrus, without affecting synapse development in CA1 pyramidal neurons (*de Wit et al., 2013*; *Roppongi et al., 2020*; *Siddiqui et al., 2013*; *Um et al., 2016*). Interestingly, we discovered now that loss of LRRTM1 and LRRTM2 in the CA1 pyramidal neurons and dentate gyrus granule cells have differential effects on synapse numbers and synaptic transmission. Outside the hippocampus, LRRTM1 is important for complex retinogeniculate synapse development and retinal convergence in visual thalamus (*Monavarfeshani et al., 2018*) and for controlling excitatory synaptic function in the mediodorsal thalamus (*Karimi et al., 2021*). Thus, LRRTMs are necessary cell-type selective mediators of synapse development and function. Additionally, NGL-2 and Latrophilin2 regulate synapse development specifically in the stratum radiatum and stratum lacunosum moleculare of CA1, respectively (*DeNardo et al., 2012*; *Sando et al., 2019*). In more general terms, synaptic specificity in the mammalian central nervous system may be conferred not by a single synapse organizer but by a set of factors that define a combinatorial code such as those contributed by cell types, subcellular compartments, and dendritic domains. Careful parsing of the roles of individual proteins or subsets of proteins will help us to devise a comprehensive catalog of the molecular factors that contribute to the development of synaptic connectivity in specific brain circuits.

Interestingly, both constitutive and adult deletion of *Lrrtm1* and *Lrrtm2* led to a robust impairment of LTP at CA3-CA1 synapses in the stratum radiatum, accompanied by a deficit in hippocampus-dependent long-term contextual memory. These results are consistent with previous KD studies (*Soler-Llavina et al., 2013*), with behavioral studies on *Lrrtm1* KO mice (*Karimi et al., 2021*; *Takashima et al., 2011*), and with the growing evidence of a key role of synapse organizing proteins in synaptic plasticity (*Bhouri et al., 2018*; *Wu et al., 2019*). Furthermore, we found that acute deletion of *Lrrtm1* and *Lrrtm2* in the dentate gyrus caused a similar LTP impairment as in the CA1 region. Although the role of LRRTM3 and LRRTM4 in synaptic plasticity remains untested, our results demonstrate that LRRTM1 and LRRTM2 contribute to LTP through mechanisms that do not require other LRRTMs.

Our results provide a causal link between the role of LRRTM1 and LRRTM2 in the control of synapse function in the dorsal hippocampus and long-term contextual memory. The region selective functions of LRRTM1 and LRRTM2 are further demonstrated by the lack of effect of *Lrrtm1* and *Lrrtm2* deletion in the dorsal hippocampus on social interaction or avoidance-related anxiety, which have been linked to the ventral hippocampus (*Fanselow and Dong, 2010*). Future studies could address the functional contributions of the LRRTMs in the ventral hippocampus and other brain regions where they are expressed.

In conclusion, the present study emphasizes the importance of studying brain-region-specific and developmental-stage-specific roles of synaptic adhesion molecules in synapse development and function. As is the case with their ligands of the neurexin family, alterations in *LRRTM* genes increase the risk for neurodevelopmental and psychiatric disorders, including autism, schizophrenia, and bipolar disorder (*Francks et al., 2007*; *Malhotra et al., 2011*; *Pinto et al., 2010*). Based on the present data, we propose that regional and temporal variations in synaptic development, transmission, and plasticity due to alterations in *LRRTM1* or *LRRTM2* contribute to these disorders.

## Materials and methods
### Generation of LRRTM1/2-DKO mice
All animal experiments complied with government and institutional guidelines. The conditional *Lrrtm1*flox/flox and *Lrrtm2* flox/flox mouse lines were described previously (*Bhouri et al., 2018*). These mice were

crossed with the EIIa-Cre line that expresses Cre recombinase in early embryonic stages (*Lakso et al., 1996*) to generate *Lrrtm1⁻/⁺* and *Lrrtm2⁻/⁺* mice. The heterozygotes were crossed to obtain single KOs. Finally, *Lrrtm1⁻/⁻* and *Lrrtm2⁻/⁻* mice were crossed to obtain LRRTM1/2-DKO mice.

## Animals

All animal experiments complied with government and institutional requirements of the University of Manitoba and conformed to ethical and procedural guidelines of the Canadian Council on Animal Care (CCAC, http://www.ccac.ca), and were approved under Protocols 15–042 and 19–054. The experimental analyses for each mice group was conducted between 6 and 7 weeks. For all experiments and analyses, the experimenter was blind to genotype.

## Tissue immunofluorescence and image analysis

Immunofluorescence studies were performed on brain tissue from 6 to 7 weeks old LRRTM1/2-DKO, P0-LRRTM1/2-cDKO and LRRTM1/2-cDKO. Mice were anaesthetized with 20% urethane and transcardially perfused with cold 0.1 M phosphate buffered saline (PBS) followed by 4% formaldehyde/ 4% sucrose in PBS (pH 7.4). The brains were then extracted, post-fixed in 4% formaldehyde overnight, and cryoprotected in 20% sucrose followed by 30% sucrose in PBS at 4 °C. For sectioning, the brains were frozen in dry ice in OCT compound (Tissue-Tek). Twenty-μm-thick coronal sections at the hippocampal level were cut in a cryostat and mounted on Superfrost Plus slides. Slides were incubated in blocking solution (5% BSA +5% normal goat serum +0.25% Triton X100 in PBS) for 1 hr, incubated at 4 °C overnight with primary antibodies (VGlut1 rabbit, 1:1000; Synaptic Systems; 135 302) and (GAD65 mouse IgG2a, 1:100, GAD-6-c, Developmental Hybridoma) diluted in the same blocking solution, followed by incubation with the appropriate secondary antibodies conjugated to Alexa 488, 568 or 647 (Invitrogen) for 1 hr at room temperature. For LRRTM1/2-DKO and P0-LRRTM1/2-cDKO analysis, DAPI (4',6 diamidino-2-phenylindole) was added to the final wash of the sections with PBS and sections were coverslipped using Immuno Mount (Thermo Scientific). For LRRTM1/2-cDKO, after final wash with PBS, sections were mounted using Fluoromount-G with DAPI (4',6 diamidino-2-phenylindole) (Southern Biotech 0100–20). In order to obtain images for synaptic density quantification, mice sections were fixed and stained simultaneously. At least three to six animals were used for each genotype.

For LRRTM1 and LRRTM2 staining, the brains were fixed and sliced according to above protocol. The sections were washed with TBSTr (50 mM Tris, pH 7.4, 1.5% NaCl and 0.3% TritonX-100) for 20 min. For immunofluorescence, the slices were then incubated in primary antibodies (LRRTM1 rabbit; Alomone Labs; ANR-141) and (anti-LRRTM2 antibody (510KSCN), described in *Linhoff et al., 2009*) diluted in TBSTr containing 10% normal goat serum for 18–24 hr at 4 °C. The sections were then washed with TBSTr for 1 hr before incubating with appropriate secondary antibodies conjugated to Alexa 568 for 1.5 hr at room temperature. After secondary antibody, the sections were washed with TBSTr for 20 min and then 50 mM Tris, pH 7.4 for 30 min. After final wash, sections were mounted using Fluoromount-G with DAPI (4',6 diamidino-2-phenylindole) (Southern Biotech 0100–20).

Imaging and image analysis was carried out blind to the experimental condition. Images were captured on a confocal microscope (Zeiss LSM 700) which is equipped for automated tiling or 63 x/1.4 NA oil objective and custom filters on a Zeiss Observer Z.1 wide-field microscope equipped with Apotome 2.0 for optical sectioning through structured illumination. For quantitative analysis, 15–20 images per mouse were acquired with a 40 x/1.4 NA oil objective at an additional ×2.5 magnification and sequential scanning with individual lasers with optimized filters. For quantification of excitatory (VGlut1) and inhibitory (GAD65) synaptic markers in brain sections, images were thresholded manually to define puncta and total integrated intensity of puncta per image area were measured. The same regions were measured for both VGlut1 and GAD65. Analysis was performed using Metamorph (Molecular Devices), Excel (Microsoft) and GraphPad Prism (GraphPad Software). Statistical comparisons were made with mutiple t-tests. All data are reported as the mean ± SEM.

## Golgi staining

To determine the density of dendritic spines, Golgi stainings of different hippocampal regions were done using the FD Rapid GolgiStain Kit (FD NeuroTechnologies), essentially as recommended by the manufacturers. Brains from euthanized 6–7 weeks old mice were rapidly removed, rinsed in PBS and

their olfactory bulbs and cerebellum excised. The rest of the brain was incubated in the rapid Golgi solutions (*Glaser and Van der Loos, 1981*) for a total of 3 weeks. The brains were then snap frozen and cut into 100-µm-thick sections at the level of the hippocampus using a cryostat. Sections were mounted on slides pre-treated with 3% gelatin. The slides were subsequently dried, dehydrated, cleared with xylene and mounted with Permount (Fisher Scientific). Imaging and image analysis was carried out blind to the experimental condition. Morphologically comparable slices from wild-type and LRRTM1/2-DKO mice were analyzed in bright field on a Zeiss Axioplan2 microscope with a 63 × 1.4 NA oil objective. Spines were counted manually on specific dendrites while altering the focal plane to account for all spines. Images of the dendrites were acquired to determine their length. Three to 4 mice were used in each group.

## Electron microscopy

Three to 4 male mice from each group (6–7 weeks old) were deeply anesthetized by intraperitoneal injection of 150 mg/kg ketamine and 15 mg/kg xylazine, and transcardially perfused with PBS followed by fixative solution (4% formaldehyde, 1.25% glutaraldehyde in PBS). The brain was removed and fixed overnight at 4 °C. A 1 mm$^3$ section of hippocampus including the CA1 region and stratum radiatum was dissected. Tissue was dehydrated in an ethanol series and embedded in JEMBED/Spurr's resin. Ultrathin sections (70 nm) were cut and stained with 2% uranylacetate and Reynold's lead and imaged on a Hitachi H7600 TEM (Hitachi). For each mouse, an equivalent region of the stratum radiatum directly underlying the CA1 pyramidal cell layer was imaged to analyze synapse density (×15,000 magnification, 50+images per mouse) or morphology (×60,000 magnification, 100+images per mouse, >500 synapses per condition). All images were acquired and analyzed blind to the genotype of each mouse.

## Immunoblotting

Three to 10 mice from each group were used for subcellular fractionation of whole brain to obtain the crude synaptosomal fraction was done essentially as described (*Roppongi et al., 2020*). For all samples, protein concentrations were normalized and run on 10% polyacrylamide gels. Gels were transferred onto Immobilon P membranes (Millipore) which were blocked in 5% skim milk in Tris-buffered saline/ 0.05% Tween-20 or 5% BSA in PBS and incubated with one of the following primary antibodies: Anti-LRRTM1 antibody (BC267) *Bhouri et al., 2018*; anti-LRRTM2 antibody (512KSCN) (*Bhouri et al., 2018*), anti-bassoon (mouse IgG2A; 1:1000; Stressgen; VAM-PS003), anti-CASK (mouse IgG1, 1:1000, K56A/50, Neuromab), anti-neuroligin 1 (mouse IgG, 1:1000, 4C12, Synaptic Systems), anti-neuroligin 1,3,4 (mouse IgG2a, 1:1000, Synaptic Systems), anti-neuroligin 3 (mouse IgG, 1:1000, N110/29, Neuromab), anti-GluN1 (mouse, 1:1000, Millipore), anti-GluN2A (rabbit, 1:1000, Chemicon), anti-pan-SAPAP (mouse IgG2b, 1:1000, N127/31, Neuromab), anti-SNAP-25 (mouse IgG1, 1:1000, Cl71.1, Synaptic Systems), anti-Synapsin (mouse IgG1, 1:1000, Synaptic Systems), anti-Synaptophysin (mouse IgG1, 1:1000, B.D. Phamigen), anti-Synaptobrevin 2 (mouse, 1:1000, Cl69.1, Synaptic Systyems), anti-Vglut1 (mouse, 1:1000, N28/9, Neuromab), anti-SynGAP (rabbit, 1:1000, Affinity Bioreagents), anti-pan-Shank (mouse IgG1, 1:1000, N23B/49), anti-TrkC (rabbit, 1:1000, C44H5, Cell Signaling), anti-LRRTM4 (mouse, 1;1000, N205B/22, Neuromab), anti-PSD-95 family (mouse IgG2a; 1:2000; clone 6G6-1C9; Millipore), anti-GluA2 (mouse; 1:1000, Millipore), anti-gephyrin (mouse IgG1; mab7, 1:1000, Synaptic Systems) and anti-βactin (rabbit, 1:5000, ab8227, Abcam) and secondary antibodies (goat anti-mouse or goat anti-rabbit HRP conjugate from Millipore). Immunoblot signals were detected using the SuperSignal Chemiluminescent kit (Thermo Scientific) and a Bio-Rad gel documentation system and normalized to the β actin levels.

## Stereotaxic injections

Animals were anesthetized with isoflurane (500–1000 ml/min flowrate) and were maintained at the surgical plane (100–200 ml/min flowrate) for the whole surgical process. After deep plane of anesthesia is obtained, the mice were administered 0.5% Bupivacaine (8 mg/kg) to intended area of incision subcutaneously and Meloxicam (2 mg/kg) subcutaneously. The mice were kept immobilized using Kopf stereotaxic apparatus. Bilateral craniotomy was made in the skull using co-ordinates for dorsal hippocampus (AP –2.1; ML ±1.4; DV –2.0). Glass pipette containing viral solution was inserted and the

virus was delivered at a flow rate of 0.1 µl/min through a picospritzer. The skin was then sutured, and the mice were allowed to recover for 3 weeks.

For P0 injections, the virus was mixed with Trypan blue to obtain a final concentration of 0.1% Trypan blue. The virus mixture was then loaded in a 5 µl Hamilton syringe. For injections, a stage was made manually according to the dimensions used previously (*Mathon et al., 2015*). The stage was then placed in a petri dish filled with ice on the stereotaxic frame. LRRTM1/2$^{floxed/floxed}$ P0 pups were anesthetized by placing the pups on ice for 2 min. Anesthesia was confirmed by gently squeezing the paws or tail of the pup and monitored lack of movement. The pup was then placed on the stage and 1 µl of the viral mixture was used to inject in both sides of the dorsal CA1 (AP –1.5; ML ±1.5; DV –1.25–1.35 from lambda). The pup's head remained held by the surgeon using an index finger for the duration of the injection. After the injection, the pups were recovered using a warming pad and returned to the home cage with the mother after the skin color returned to pink and the pup regained movement. The pups were monitored after the injection for 72 hr to check for health and visible milk spot.

## Electrophysiology

For preparation of mouse hippocampal slices, 6- to 7-week-old LRRTM1/2-DKO, wild-type littermates, LRRTM1/2-cDKO, P0 LRRTM1/2-cDKO and control injected mice underwent cervical dislocation and decapitation. Four to 5 mice were used in each group with 3–4 slice recordings from each animal.

### Field recordings

For LRRTM1/2-DKO LTP analysis, whole brains were transferred to ice-cold slicing solution consisting of (in mM): 120 NMDG, 2.5 KCl, 1.2 NaH2PO4, 25 NaHCO3, 1.0 CaCl2, 7.0 MgCl2, 2.4 Na-pyruvate, 1.3 Na-ascorbate, 20 D-glucose with pH adjusted to 7.35 using HCl acid (unless stated, all chemicals and drugs were purchased from Sigma or BioShop, Canada). The hippocampus was removed and sliced in the transverse plane using a manual tissue chopper (Stoelting, Wood Dale, IL, USA). Slices were maintained in a recovery chamber at 30 °C for 1 hr in ACSF composed of the following: (in mM): 124 NaCl, 3 KCl, 1.25 NaH2PO4, 1 MgSO47H2O, 2 CaCl2, 26 NaHCO3 and 15 D-glucose which was bubbled continuously with carbogen (95% O2/5% CO2) to adjust the pH to 7.3. After an additional 30 min at room temperature, slices were transferred to a submerged recording chamber and were continually perfused with ACSF (2–3 ml/min) at room temperature. Extracellular field EPSPs (fEPSPs) were elicited by using two bipolar nickel–chromium electrodes placed in stratum radiatum of area CA1 and recorded with a glass microelectrode filled with ACSF (resistances, 2–3 MΩ). Baseline responses were set at 40% of maximum and taken for 20 min prior to inducing LTP. High-frequency stimulation consisting of 1 × 100 Hz, 1 s duration was used to induce LTP. Extracellular recordings were acquired using WinLTP. The initial slope of the fEPSP was measured to quantify synaptic strength (*Johnston, 1995*). Student's t-test was used for statistical comparisons of mean fEPSP slopes between groups. All values shown are mean ± SEM with n=number of slices.

For LRRTM1/2-cDKO slice electrophysiology, mice were anesthetized and the brain was rapidly removed and placed in ice-cold standard ACSF composed of (in mM): 124 NaCl, 3 KCl, 1.25 NaH2PO4, 1.3 MgSO47H2O, 2.6 CaCl2, 26 NaHCO3 and 10 D-glucose saturated with 95% O2 and 5% CO2. The hippocampus was removed and sliced in the transverse plane (350 µm thickness) using vibratome (ThermoFisher vibrating microtome 650 V). Slices were transferred in ACSF and incubated for 1 hr at room temperature to equilibrate before placing in a recording chamber continuously perfused with 95% O$_2$ and 5% CO$_2$ at a flow rate of 3 mL/minute (for CA1) and 1.5 mL/min (for dentate gyrus). For CA1, 2 µM bicuculline methiodide was added to standard ACSF. Recording electrodes consisted of borosilicate pipettes (TW150-F3; WPI, Sarasota, FL, USA) pulled with a two-stage puller (PP-83; Narishige, Greenvale, NY, USA). Extracellular field EPSPs (fEPSPs) were elicited by using two electrodes placed in stratum radiatum of area CA1 and recorded with a glass microelectrode filled with ACSF (resistances, 2–3 MΩ). Baseline responses were set at 30–40% of maximum and taken for 20 min prior to inducing LTP. High-frequency stimulation consisting of 2 × 100 Hz, 1 s duration was used to induce LTP in CA1. Student's t-test was used for statistical comparisons of mean fEPSP slopes between groups. All values shown are mean ± SEM, with n=number of slices.

For DG LTP, 10 uM bicuculline methiodide was added to standard ACSF. For recording fEPSPs, the stimulating electrode was positioned on dentate gyrus molecular layer in 10–12 µm away from

granular layer (inner molecular layer) and responses were recorded using a borosilicate glass capillary pipette filled with ACSF with a tip resistance of 2–3 MΩ. High-frequency stimulation consisting of 4 × 100 Hz, 1 s duration was used to induce LTP in dentate gyrus. Student's t-test was used for statistical comparisons of mean fEPSP slopes between groups. All values shown are mean ± SEM, with n=number of slices.

Paired pulse ratio was recorded from CA1 of Global DKO (LRRTM1/2-DKO) and Local late cDKO (LRRTM1/2-cDKO) CA1 at stimulation intensities that resulted in response-amplitudes 1/3 of maximum. Inter-stimulus intervals were as follows:, 20ms, 50ms, 100ms, 250ms, and 500ms.

### Patch clamp recordings

Slices for whole-cell recordings of LRRTM1/2-DKO CA1, brain slices were prepared as described above. Neurons were recorded using the "blind" method with a MultiClamp 700B amplifier with cells voltage clamped at –70 mV and continually perfused with ACSF (same as above). Recording pipettes were filled with solution containing (in mM): 122.5 Cs-methanesulfonate, 17.5 CsCl, 2 MgCl2, 10 EGTA, 10 HEPES, 4 ATP(K), and 5 QX-314, with pH adjusted to 7.2 by CsOH. Prior to recording mEPSCs, bicuculline methiodide (10 µM; Abcam) and tetrodotoxin (TTX; 500 nM; Ascent Scientific) were added to block GABA receptor-mediated inhibitory synaptic currents, and action potentials respectively. CNQX (10 µM; Abcam) and DL-AP5 (50 µM; Abcam) were added toward the conclusion of some experiments to confirm that remaining currents were mEPSCs. mEPSCs were recorded using WinLTP in continuous acquisition mode. Analyses for frequency and amplitude were conducted using MiniAnalysis software. Statistical analyses were completed using GraphPad InStat and SigmaPlot. Student's t-tests were conducted to test for differences between groups with statistical significance set at $p<0.05$ with n=number of cells. Data are presented as mean ± SEM.

For all other whole-cell recordings (LRRTM1/2-DKO DG, LRRTM1/2-cDKO and P0 LRRTM1/2-cDKO), mice were anesthetized and the brain was rapidly removed and placed in ice-cold standard ACSF composed of (in mM): 124 NaCl, 3 KCl, 1.25 NaH2PO4, 1.3 MgSO47H2O, 2.6 CaCl2, 26 NaHCO3, 10 D-glucose and 3 Na-Pyruvate saturated with 95% O2 and 5% CO2. The hippocampus was removed and sliced in the transverse plane (250 µm thickness) using vibratome (Campden model 7,000smz-2). Slices were then recovered in NMDG ACSF for 5 min at 32 °C. NMDG ACSF was composed of (in mM): 94 NMDG, 3 KCl, 1.25 NaH2PO4, 30 NaHCO3, 0.5 CaCl2, 5 MgCl2, 3 Na-pyruvate, 5 Na-ascorbate, 25 D-glucose with pH adjusted to 7.35 using HCl acid. The slices were transferred to HEPES ACSF and recovered for 1 hr at room temperature before recording. HEPES ACSF was composed of (in mM): 95 NaCl, 3 KCl, 1.25 NaH2PO4, 30 NaHCO3, 0.5 CaCl2, 5 MgCl2, 3 Na-pyruvate, 5 Na-ascorbate, 25 D-glucose with pH adjusted to 7.35. Recording pipettes were filled with solution containing (in mM): 142 cesium gluconate, 10 HEPES, 8 NaCl, and 2 MgCl2, pH 7.2 (adjusted with 1M-CsOH). Prior to recording mEPSCs, tetrodotoxin (TTX; 100 nM; Alomone lab) were added to block action potentials. mEPSCs were recorded using Clampex software. The cells were voltage clamped at –60 mV and mEPSCs were recorded for a minimum period of 2–5 min. Student's t-tests were conducted to test for differences between groups with statistical significance set at $p<0.05$ with n=number of cells. Data are presented as mean ± SEM.

## Contextual fear conditioning

Contextual fear conditioning (CFC) for LRRTM1/2-DKO mice was performed in a black Plexiglas chamber with a camera mounted above for recording and offline behavioral analysis. Mice were gently handled for 3 days prior to experiments. The mice were habituated to the context by placing them in the chamber for 2 min prior to administering a single electrical foot shock (0.7 mA foot shock, 2 s duration) administered through a stainless-steel grid floor. Following conditioning, mice remained in the chamber for an additional 3 min to allow for encoding of the context +shock association. Mice were subsequently returned to their home cage after completing the conditioning procedure. CFC was then assessed 24 hr after the conditioning session by placing mice in the conditioning chamber for 5 min. Absence of movement except respiration constituted freezing time which was converted to freezing percentage (total amount of freezing time/5 min) to assess contextual memory for each mouse. Student's t-tests was performed to compare the amount of freezing between groups.

For LRRTM1/2-cDKO, the mice were habituated to the context by placing them in the chamber for 10 min on the first day. Twenty-four hr later, three electrical foot shocks (1 mA foot shock, 30 s

interval) were administered through a stainless-steel grid floor after placing the mice in chamber for 2 min. Following conditioning, mice remained in the chamber for an additional 1 min to allow for encoding of the context +shock association. Mice were subsequently returned to their home cage after completing the conditioning procedure. CFC was then assessed 24 hr after the conditioning session by placing mice in the conditioning chamber for 5 min. Absence of movement except respiration constituted freezing time which was converted to freezing percentage (total amount of freezing time/5 min) to assess contextual memory for each mouse. Rearing was analyzed as the time assessed when each mouse would keep two front paws on the wall. Analysis were done on AnyMaze software. Student's t-tests was performed to compare the amount of freezing between groups. Ten to –11 mice were used in each group.

### Accelerating rotarod

An accelerating rotarod (Harvard Apparatus, USA) was used to assess gross motor skills and motor coordination of mice. Mice were brought to the behavior room 10 min before starting the tests. The rotarod was set to 4 rpm, accelerating rate 20 rpm/min. The mice were placed on the rotarod opposite from the direction of acceleration. Three trials were given to each mouse. Time to fall from the rotarod (latency to fall) was measured for each mouse. Average of the three trials were included in analysis. Student's t-tests was performed to compare between groups. Ten to 11 mice were used in each group.

### Open-field maze

The open-field maze was made of 50 cm (length) x 50 cm (width) x 38 cm (height) non-porous hard plastic. The mice were brought to the behavior room 20 min before the start of the test. Each mouse was placed in the center of the maze and allowed to explore the maze for 10 min. After 10 min, the mice were returned to their home cages. The behavior was then analyzed using AnyMaze software. Student's t-tests was performed to compare between groups. Ten to 11 mice were used in each group.

### Elevated plus maze

The maze is made of hard plastic of four arms with dimensions 30 cm long and 5 cm wide (two open without walls and two enclosed by 15.25 cm high walls). The maze is elevated 40 cm off the table with legs. The mice were brought 15 min before the behavioral test for acclimatization. Each test mouse is placed in the center of the maze with its head facing the closed arms and opposite to the experimenter. The mice were allowed to explore the maze for 10 min before putting back in their home cages. The behavior was then analyzed using AnyMaze software. Student's t-tests was performed to compare between groups. 10–11 mice were used in each group.

### Crawley's three-chambered social approach

The mice were brought to the behavior room 20 min before the start of the test to acclimate. The sociability chamber comprised of a rectangular box with three chambers of equal size (20 × 40 x 22 cm) made from clear Plexiglas. The test mouse was introduced in the middle chamber of three-chambered apparatus for 10 min with doors closed to both chambers for habituation. After 10 min, a Stranger I mouse (the placement of Stranger I in the left or right side of the chamber was randomly altered between trials) was placed in one of the inverted wire cups. The doors of the chambers were opened, and the test mouse is allowed to explore both the chambers (stranger I and empty wire cup). Ten min later, Stranger II mouse is placed in the empty wire cup for another 10 min. Time spent by the test mice in each chamber was measured and analyzed manually with experimenter being blind to groups. The apparatus was cleaned with 10% ethanol after every session. Note: All Stranger mice were of the same gender as the subject mice and had no prior interaction with them. Ten to 11 mice were used in each group.

### Acknowledgements

We thank Nazarine Fernandes for excellent technical assistance. This work was supported by NSERC (RGPIN-2015–05994) and Research Manitoba New Investigator Grant (to TJS), Research Manitoba studentship and Rady FHS Scholarship (to SD), CIHR grants (MOP-142209 to TJS, MOP-130526 to SXB, FND-154286 to YTW, and FDN-143206, MOP-84241 to AMC, and PJT-173550 to MFJ), Canada

Research Chairs program (to SAC), JSPS KAKENHI Grant Numbers JP15K21769 and JP20K07334, Ohsumi Frontier Science Foundation, Takeda Science Foundation and The Uehara Memorial Foundation (to HK). YTW is the holder of the Heart and Stroke Foundation of British Columbia and Yukon Chair in Stroke Research.

## Additional information

### Competing interests

Nils Brose: Reviewing editor, *eLife*. The other authors declare that no competing interests exist.

### Funding

| Funder | Grant reference number | Author |
|---|---|---|
| Natural Sciences and Engineering Research Council of Canada | RGPPIN-2015-05994 | Tabrez J Siddiqui |
| Research Manitoba | New Investigator | Tabrez J Siddiqui |
| Research Manitoba | Studentship | Shreya H Dhume |
| Rady Faculty of Health Sciences, University of Manitoba | Scholarship | Shreya H Dhume |
| Canadian Institutes of Health Research | MOP-130526 | Shernaz X Bamji |
| Canadian Institutes of Health Research | FND-154286 | Yu Tian Wang |
| Canadian Institutes of Health Research | FDN-143206 | Ann Marie Craig |
| Canadian Institutes of Health Research | MOP-84241 | Ann Marie Craig |
| Canada Research Chairs | | Steven A Connor |
| JSPS KAKENHI | JP15K21769 | Hiroshi Kawabe |
| JSPS KAKENHI | JP20K07334 | Hiroshi Kawabe |
| Ohsumi Frontier Science Foundation | | Hiroshi Kawabe |
| Takeda Science Foundation | | Hiroshi Kawabe |
| Uehara Memorial Foundation | | Hiroshi Kawabe |
| Heart and Stroke Foundation of British Columbia and Yukon | Chair in Stroke Research | Yu Tian Wang |
| Canadian Institutes of Health Research | MOP-142209 | Tabrez J Siddiqui |
| Canadian Institutes of Health Research | PJT-173550 | Michael F Jackson |

The funders had no role in study design, data collection and interpretation, or the decision to submit the work for publication.

### Author contributions

Shreya H Dhume, Formal analysis, Investigation, Methodology, Validation, Visualization, Writing – original draft, Writing – review and editing; Steven A Connor, Formal analysis, Investigation, Methodology, Supervision, Validation, Visualization; Fergil Mills, Formal analysis, Investigation, Validation,

Visualization; Parisa Karimi Tari, Formal analysis, Investigation; Sarah HM Au-Yeung, Investigation, Validation, Visualization; Benjamin Karimi, Shinichiro Oku, Reiko T Roppongi, Investigation; Hiroshi Kawabe, Shernaz X Bamji, Supervision; Yu Tian Wang, Resources, Supervision; Nils Brose, Resources, Supervision, Writing – original draft, Writing – review and editing; Michael F Jackson, Resources, Supervision, Writing – review and editing; Ann Marie Craig, Conceptualization, Funding acquisition, Methodology, Project administration, Resources, Supervision, Writing – original draft, Writing – review and editing; Tabrez J Siddiqui, Conceptualization, Formal analysis, Funding acquisition, Investigation, Methodology, Project administration, Resources, Supervision, Validation, Visualization, Writing – original draft, Writing – review and editing

### Author ORCIDs
Shinichiro Oku http://orcid.org/0000-0002-1916-1870
Hiroshi Kawabe http://orcid.org/0000-0001-5650-8696
Shernaz X Bamji http://orcid.org/0000-0003-0102-9297
Yu Tian Wang http://orcid.org/0000-0001-8592-0698
Ann Marie Craig http://orcid.org/0000-0002-8651-8200
Tabrez J Siddiqui http://orcid.org/0000-0002-6938-7827

### Ethics
All animal experiments complied with government and institutional requirements of the Universities of British Columbia and Manitoba and conformed to ethical and procedural guidelines of the Canadian Council on Animal Care (CCAC, http://www.ccac.ca), and approved in Protcols 15-042 and 19-054.

### Decision letter and Author response
Decision letter https://doi.org/10.7554/eLife.64742.sa1
Author response https://doi.org/10.7554/eLife.64742.sa2

## Additional files

### Supplementary files
• Transparent reporting form

### Data availability
All data generated during this study are included in the manuscript and supporting files. Source data files have been provided for all main Figures (Figures 1-9), and for Figure 1 - Figure supplements 1,2,3, Figure 4- Figure Supplement 1, Figure 6- Figure supplement 2, Figure 8 - Figure supplement 1, and Figure 9- Figure supplement 1.

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
