## [Editor Report]

This study aims to systematically investigate the role of LRRTM 1 and 2 in hippocampal circuit function. These neuronal proteins are organizing proteins in the synapse and are known to play a role in development and plasticity. The current literature is unclear on the exact role these proteins play due to the diverse technical approaches used in previous studies which often led to contradictory findings. The current study aims to clearly identify the roles of these proteins by using a thorough experimental design. The clear strength of the study is in the systematic and careful comparison between various approaches that aim to eliminate LRRTM1/2 from the synapses and the thorough examination of several parameters under these different conditions.

---

## [Decision Letter]

**Decision letter after peer review:**

Thank you for submitting your article "Distinct but overlapping roles of LRRTM1 and LRRTM2 in developing and mature hippocampal circuits" for consideration by *eLife*. Your article has been reviewed by 3 peer reviewers, and the evaluation has been overseen by a Reviewing Editor and John Huguenard as the Senior Editor. The reviewers have opted to remain anonymous.

The reviewers have discussed the reviews with one another and the Reviewing Editor has drafted this decision to help you prepare a revised submission.

Summary:

The current study aims to systematically investigate the role of LRRTM 1 and 2 in hippocampal circuit function. These neuronal proteins are organizing proteins in the synapse and known to play a role on development and plasticity. The current literature in unclear on the exact role these proteins play due to the diverse technical approaches used in previous studies which often led to contradictory findings. The current study aims to clearly identify the roles of these proteins by using a thorough experimental design. The reviewers were positive about the prospect of the current study offering clarity on previously disputed questions. They found that the data quality is high, and the mains conclusions are well supported, and this work is important contribution to the field. At the same time, it was noted that this study does not offer substantially novel functional insights into the mechanism by which LRRTM1/2 contributes to synaptic functions. In the absence of these novel insights, the clear strength of the study is in the systematic and careful comparison between various approaches that aim to eliminate LRRTM1/2 from the synapses and the thorough examination of several parameters under these different conditions. For this reason, the authors need to significantly improve this angle of their study so that it can stand as a critical work that sheds light on previously contradictory findings. The reviewers identified several key areas that would need to be implemented with additional data. These are listed below and the rationale for these experiments is explained in the reviewer's individual reports.

Essential revisions:

1. Linking the 2 parts of the study together through a more consistent experimental design that allows the authors to property differentiate between the effects of local vs global and developmental changes in LRRTM1/2 levels.

2. Assessment of KD with WB or IHC.

3. Consistent parallel measurements from DG and CA1.

4. Evaluation of presynaptic function, mini analysis from cDKO CA neurons.

5. Synapse density from all dendritic layers in the DG.

*Reviewer #1:*

Important take aways from this study include: (1) the global loss of LRRTM1/2 results in a ~30% loss of excitatory synapses in the CA1 region of hippocampus; (2) the global loss of LRRTM1/2 results in impaired LTP and contextual fear conditioning; (3) the postnatal (post developmental) loss of LRRTM1/2 does not lead to a loss of synapses in CA1 but does impair LTP and contextual fear conditioning.

This is an important contribution to the field.

However, I do have a few concerns.

Most importantly, the link between the two halves of the study (global LRRTM1/2 loss versus temporal/regional deletions) could be strengthened. As it stands now, distinct methods were used to generate these two models of LRRTM loss and that could lead to over interpretation of the results. In the first half of the paper LRRTM1/2 are lost globally which means that they are lost from both the pre- and postsynaptic partners in CA1 (as well as everywhere else in the brain). In the second half, the loss of LRRTM1/2 is driven by regional, viral administration of Cre into dorsal hippocampus after developmental synaptogenesis. One interpretation is that the loss of these adhesion molecules after developmental synaptogenesis does not impair synapse number as the developmental roles for these molecules have already passed. However, an alternative is that the regional delivery is not removing LRRTMs from the necessary cells. While LRRTMs are well established for roles in postsynaptic cells, it remains possible that they have yet to be defined non-synaptic or even presynaptic roles. To better compare the phenotypes in the first and second halves of this study I see two approaches. One could be to use a global CreER rather than viral delivery. I understand what a pain that would be and how much time it would take. The alternative could be to use viral administration of Cre into the P0 dorsal hippocampus. Based on how they describe the turnover of LRRTMs this should delete these adhesion molecules well before synaptogenesis (and well before there analysis which was at 6 weeks) and could better link the two halves of the study.

Another concern that I had involves the assessment of KD of LRRTMs after viral administration of Cre. The authors cite Fornasiero et al. 2018 that the turnover of LRRTMs is quite rapid. While I could not find this in the manuscript, I assume it is in the data deposited elsewhere. Its actually quite a bit faster than the turnover for most of the other synaptic cell adhesion molecules studied in that paper. For this reason, it seems critical to assess the knock down of LRRTMs (by WB or IHC), not just by assessing the number of cells infected. This seems especially important because one of the takehome messages from Fornasiero et al. 2018 was that different brain regions and different synaptic molecules have unique half lives.

A concern with the methods for IHC analysis involved how images were processed before they were quantified. The authors state that "images were thresholded manually". So similar thresholds were not used for images that were compared (i.e. control versus dKO)? How were such thresholds selected? Did the authors assess whether varying the thresholds impacted the values of the quantified data? There is data out there that manual thresholding can impact values.

*Reviewer #2:*

The authors' key conclusion is that the LRRTM1 and 2 molecules play different roles in different places at different times. This is an important conclusion, though most of the observations here are not very surprising based on the existing literature.

With early and widespread knockout, synapse numbers (spines, PSDs, mini frequency) are reduced in CA1 but not dentate gyrus. This is not surprising given that we know LRRTM3/4 play important roles in DG and thus are available to compensate there, but are not expressed significantly in CA1.

– It is confusing to describe the DKO mice from start to finish in the methods without reference to the previous cDKO mice. My impression is that the mice line used in Bhouri 2018 was crossed with Ella-Cre mice here. Is that correct, or were they regenerated with the same tools as the mice described in Bhouri 2018?

– Parallel measurements in DG and CA1 would strengthen the argument that this paper provides the definitive demonstration of unique and separable roles in the two cell types. Importantly, excluding all detail from analysis of DG synapses in the cDKO animals not only provides a stark contrast with the extensive analysis in the DKO animals but will make for a difficult time interpreting the effects on LTP. Ideally, the analyses in Figure 1S2, 2, 3, and 4 would be carried out in granule cells and their perforant path synapses, though rather than repeating this battery to search for negative effects, it might be sufficient merely to provide a more specific measure of synapse number and a representative measure of synapse function.

– The effect on post-tetanic potentiation following the stimulus appears strikingly different in Figure 4A and 6B, perhaps suggestive of differential presynaptic effects during the tetanic stimulus, but no assessment of presynaptic function is provided.

– The LTP experiments in Figure 4 begin with an unbalanced synaptic drive. If 20% of the synapses are lacking, then delivering the stimulus at the same proportion of the maximal response will also be presumably driving the cells with 20% less input. This could cause a difference in the magnitude of LTP. The comparison thus appears a bit unfair, particularly without any assessment of presynaptic function e.g. during the train induction where the remaining DKO synapses could have more quickly decrementing release. (The authors may point out that no difference in PTP magnitude in Figure 4 indirectly argues against this possible difference in presynaptic function.)

There is further no attempt at discerning a behavioral role for the LTP deficit in DG.

Some clarification on staining and synapse counting would be useful. It seems that the only analysis of synapse numbers with the late cKO is VGlut or VGAT staining. Are AMPARs lost from the remaining synapses in the early or late DKO neurons?

*Reviewer #3:*

Dhume and colleagues investigate the contribution of the postsynaptic adhesion molecules LRRTM1 and LRRTM2 to hippocampal circuit function. By crossing floxed LRRTM1 and LRRTM2 mice with the EIIa Cre line, they generate double knockout (DKO) mice. By broadly infecting floxed LRRTM1 and LRRTM2 mice with Cre recombinase expressing AAV, they generate conditional DKO (cDKO) mice. These strategies allow the authors to compare the effects of chronic LRRTM1/2 deletion in DKO mice vs. the effects of LRRTM1/2 deletion in cDKO mice after synapse development has largely completed. The authors find reduced excitatory synapse density in the CA1 region of LRRTM1/2 DKO mice but not cDKO mice. LTP is impaired in both DKO and cDKO mice, confirming previous results (Bhouri et al., PNAS 2018). Contextual fear conditioning is impaired in both DKO and cDKO mice.

The experiments in the paper are well-performed and clearly presented, and provide additional insight in the role of LRRTM1/2 in circuit development and function. The study is a follow up to the previously published work by Bhouri et al., but uses two different deletion strategies and adds data on synapse density and behavioral paradigms. The LTP experiments presented are similar to that study, with the difference that Bhouri et al. recorded LTP in sparsely infected CA1 neurons from LRRTM1/2 cDKO mice compared to the population measurements in this study, and the additional analysis of LTP in cDKO DG in this study.

1) Figure 1 —figure supplement 1 panels B and C appear to show western blot analysis of LRRTM1 and LRRTM2 levels in different LRRTM1 and LRRTM2 genotypes, but the figure legend mentions PCR. Please clarify.

If this is western blot as the text, blot and markers in kDa suggest, which LRRTM antibodies were used? I could not find this information in the Methods section but may have missed it. The bands detected that appear to represent LRRTM1 and LRRTM2 run low, at or below the predicted molecular weight of 58 kDa (uniprot) for these proteins, which seems low for glycosylated membrane proteins. The band pattern also appears different from what was reported before in Bhouri et al., PNAS 2018. Please explain.

2) Figure 1B, C: what was the rationale for only analyzing synapse density in the medial molecular layer in DG? Linhoff et al., Neuron 2009 showed prominent LRRTM2 immunoreactivity in the outer molecular layer of the DG. Analyzing synapse density in all dendritic layers in DG should be performed, also to allow comparison with the data presented in Figure 5 (analysis of synapse density in all layers of DG in LRRTM1/2 cDKO).

3) The cDKO strategy should be validated, using western blot analysis of dorsal hippocampal lysate.

4) Using the cDKO strategy but sparser deletion of LRRTM1/2, Bhouri et al. showed a decreased mEPSC amplitude (but not frequency) in CA1 neurons. It would be informative to analyze mEPSC frequency and amplitude in LRRTM1/2 cDKO CA1 neurons in this study, to compare to the synapse density data shown in Figure 5.

5) As a suggestion, the Results section might benefit from a more explicit comparison in the text between the two strategies used by the authors (i.e. chronic LRRTM1/2 deletion in Figures 1-4 vs broad viral vector-mediated deletion of LRRTM1/2 in 3 week-old animals after synapse development has completed in Figure 5-7). Now the text for the cDKO mice section emphasizes analysis in the mature brain (page 11), whereas the timepoint of analysis (injection in 3 week-old mice + 3 weeks for analysis) does not differ much from the DKO mice (6-7 weeks old). As another example, the description of the contextual fear memory experiment in Figure 7 is more elaborate than the description for (what appears to be) the same experiment in Figure 4. The findings from the two genetic strategies could be contrasted more clearly, and in some cases, such as the fear memory experiment, even be presented side-by-side for example.

Experiments to address my points 2 and 3 are needed to strengthen the study.

---

## [Author Response]

Essential revisions:1. Linking the 2 parts of the study together through a more consistent experimental design that allows the authors to property differentiate between the effects of local vs global and developmental changes in LRRTM1/2 levels.

The original study had two experimental models: (1) global deletion of LRRTM1 and 2 from birth and analysis at 6-7 weeks of age (Global DKO); and (2) conditional deletion of LRRTM1 and 2 in the dorsal hippocampus and analysis at 6-7 weeks of age (Local late cDKO). For a more consistent study design and to properly differentiate between the effects of local vs global and developmental changes in LRRTM1 and 2 levels, we added several new experiments. The new experiments are now included in Figures 1, 3, 5, 7, Figure 1—figure supplement 3, Figure 4—figure supplement 1, Figure 5—figure supplement 1, Figure 6—figure supplement 1, Figure 8—figure supplement 1.

To better link the two parts of the study, we analysed a third model – deletion of LRRTM1 and 2 in the CA1 at postnatal day 0 (P0) and analysis at 6-7 weeks of age. Thus, the revised manuscript now includes three distinct deletion models to comprehensively understand the contributions of LRRTM1 and 2 to synapse development and function: (1) *Global DKO*, (2) *Local early cDKO (P0),* and (3) *Local late cDKO*. For consistency, all analyses were done at 6-7 weeks of age. This design allowed us to cleanly and comprehensively differentiate between the effects of local versus global knockout and early versus late knockout. The new experiments strengthened our original conclusions about the context-dependent functions of LRRTM1 and 2 in the CA1 where LRRTM1 and LRRTM2 are the only family members expressed and, in the DG, where all LRRTM family members (LRRTM1-4) are expressed. The summary of our findings are as follows.

i) In the CA1 of Global DKO, excitatory synapse numbers were reduced in all layers of the CA1 as assessed by reduced VGlut1 synaptic puncta immunofluorescence (Figure 1B and C). Accordingly, ultrastructural studies revealed reduced excitatory synapse numbers and reduced spine density in Global DKO (Figure 2). Consistent with these findings, mEPSC frequency was reduced but mEPSC amplitude was unaffected (Figure 3A-C).

As advised by the reviewers, the Global DKO model does not allow us to distinguish whether the effects of *Lrrtm1* and *2* deletion is pre- or postsynaptic. Therefore, we developed and assessed synaptic function in a new deletion model – the *Local early cDKO (P0)* model. Consistent with the Global DKO, we found that local early deletion of *Lrrtm1* and *2* in the CA1 significantly reduced mEPSC frequency (Figure 5). A modest reduction in the mEPSC amplitude was also observed. These results indicate that LRRTM1 and 2 mediate postsynaptic functions in the CA1. The *Local early cDKO (P0)* model did not, however, allow us to reliably measure VGluT1 and GAD65 synaptic puncta immunofluorescence because to selectively target the CA1, we could only obtain partial infection of CA1 using a reduced viral load. Increasing the viral load could target more CA1 neurons but caused spillover in the CA3 and dentate gyrus, and would thus not be an appropriate experimental design. Therefore, in the *Local early cDKO (P0)* model, we focused on mEPSCs from individual infected neurons, which is a reliable measure of synaptic function.

In the CA1 of Local late cDKO, VGlut1 synaptic puncta immunofluorescence was indistinguishable from that in control mice (Figure 6D and E). Accordingly, mEPSC frequency was unaltered. Consistent with our previous report (Bhouri et al., 2018), mEPSC amplitude was reduced in the CA1 of Local late cDKO (Figure 7A-C).

ii) LTP deficits are apparent in both Global DKO (Figure 4A and B) and Local late cDKO (Figure 8). However, the locus of these deficits appears to be mechanistically diverse. Impairments in LTP in Global DKOs are restricted to the maintenance phase (Figure 4A and B), suggesting that presynaptic plasticity in response to high frequency stimulation is intact. We observed modest change in paired-pulse facilitation, limited to delayed interstimulus intervals (ISIs) of 100 and 200 ms, suggesting minimal consequence of Global DKO on presynaptic short-term plasticity. The locus of the deficits is likely postsynaptic, due to impaired stabilization of synaptic AMPARs in the absence of LRRTMs, as we have shown earlier (Bhouri *et al.*, 2018). In contrast, the Local late cDKOs demonstrate marked reductions in the extent of post-tetanic potentiation following HFS (Figure 8). Note, that a similar PTP deficit in response to late cDKO has been reported previously (Bhouri *et al.*, 2018). As suggested by Reviewer 2, we have assessed pre-synaptic function by examining paired-pulse facilitation (Figure 8 —figure supplement 1). We now show that there is no difference between control and Local late cDKOs suggesting that mechanisms underlying shortterm facilitation are not impacted by Local late cDKO. We speculate that presynaptic mechanisms distinct from the paired-pulse facilitation may underlie the PTP deficits in the Local late cDKO. Alternatively, the complement of postsynaptic glutamatergic receptors or their retention in the postsynaptic membrane following LTP may be altered in the absence of LRRTM1/2, shifting the induction threshold for LTP, as we have shown earlier (Bhouri et al., 2018). Investigations into these possibilities are beyond the scope of the current manuscript but could be pursued in future studies.

iii) In the DG, effects of Local late cDKO of LRRTM1 and 2 was similar to that observed in the CA1 – no changes in VGlut1 puncta immunofluorescence (Figure 6F and G) or mEPSC frequency but reduced mEPSC amplitude (Figure 7D-F) and impaired LTP (Figure 8D-F). These results suggest that in the DG, LRRTM1 and 2 contribute to synaptic transmission and LTP through mechanisms that are distinct from that of LRRTM3 and LRRTM4, that are co-expressed in the DG. However, in the DG of Global DKO, whereas VGlut1 puncta immunofluorescence was unchanged in the inner and medial molecular layers, it was increased in the outer molecular layer (Figure 1D and E), and was accompanied by increased mEPSC frequency (Figure 3D-F). These results point to a possible compensatory mechanism within this circuit, by other dentate gyrus resident synapse organizers, requiring further investigation in the future.

2. Assessment of KD with WB or IHC.

As advised, we confirmed effective depletion of LRRTM1 and LRRTM2 in the Local late cDKO using immunocytochemistry of brain sections (Figure 6—figure supplement 1). LRRTM1 is enriched in the stratum radiatum whereas LRRTM2 is enriched in the stratum lacunosum moleculare of CA1 and outer molecular layer of the dentate gyrus (Figure 6—figure supplement 1). AAV-Cre delivery selectively to the CA1 of LRRTM1/2 floxed mice depleted LRRTM1 and LRRTM2 immunofluorescence intensity in the CA1 but not in the DG. Thus, our approach using viral methods was effective in ablating LRRTM1 and LRRTM2 in targeted regions.

3. Consistent parallel measurements from DG and CA1.

For consistent parallel measurements from DG and CA1, we added several new experiments for a total of nine parallel measurements. Thus, as advised by the reviewers, we assessed synapse numbers, function and plasticity in both CA1 and DG of Global DKO and Local late cDKO. The nine parallel measures include:

1. VGlut1 puncta immunofluorescence in all dendritic layers in Global DKO – CA1 (Figure 1B and C) and DG (Figure 1D and E)

2. VGlut1 puncta immunofluorescence in all dendritic layers in Local late cDKO – CA1 (Figure 6D and E) and DG (Figure 6F and G)

3. GAD65 puncta immunofluorescence in all dendritic layers in Global DKO – CA1 (Figure1 —figure supplement 2A and B) and DG (Figure1 —figure supplement 2C and D)

4. GAD65 puncta immunofluorescence in all dendritic layers in Local late cDKO – CA1 (Figure6 —figure supplement 2A and B) and DG (Figure6 —figure supplement 2C and D)

5. miniature EPSC frequency in Global DKO – CA1 (Figure 3B) and DG (Figure 3E)

6. miniature EPSC amplitude in Global DKO – CA1 (Figure 3C) and DG (Figure 3F)

7. miniature EPSC frequency in Local late cDKO – CA1 (Figure 7B) and DG (Figure 7E)

8. miniature EPSC amplitude in Local late cDKO – CA1 (Figure 7C) and DG (Figure 7F)

9. Long-term potentiation in Local late cDKO – CA1 (Figure 8A-C) and DG (Figure 8D-F)

4. Evaluation of presynaptic function, mini analysis from cDKO CA neurons.

As discussed above (*Essential Revision 1*), we assessed paired-pulse ratios in CA1 neurons for both Global DKO and Local late cDKO mice. No change in paired pulse facilitation was observed in Local late cDKOs when compared to controls (Figure 8 —figure supplement 1). Additionally, we observed change in paired pulse facilitation in Global DKO at delayed (100 and 250 ms), but not early (20 and 50 ms), interstimulus intervals (Figure 4—figure supplement 1). Further, as advised, we completed mini analysis from LRRTM1/2-cDKO CA1 neurons (Figure 7A-C), and to strengthen the study, we also did parallel mEPSC measurements from LRRTM1/2-cDKO DG granule cells (Figure 7D-F).

5. Synapse density from all dendritic layers in the DG.

We performed confocal analysis for VGlut1 and GAD65 synaptic markers in all dendritic layers of dentate gyrus in Global DKO mice, which is now included in the revised manuscript (Figure 1D and E; Figure 1—figure supplement 3C and D). We had already included confocal analysis from all dendritic layers of dentate gyrus in Local late cDKO mice in the original submission (Figure 6F and G; Figure 6 —figure supplement 2C and D).

Reviewer #1:Important take aways from this study include: (1) the global loss of LRRTM1/2 results in a ~30% loss of excitatory synapses in the CA1 region of hippocampus; (2) the global loss of LRRTM1/2 results in impaired LTP and contextual fear conditioning; (3) the postnatal (post developmental) loss of LRRTM1/2 does not lead to a loss of synapses in CA1 but does impair LTP and contextual fear conditioning.This is an important contribution to the field.A concern with the methods for IHC analysis involved how images were processed before they were quantified. The authors state that "images were thresholded manually". So similar thresholds were not used for images that were compared (i.e. control versus dKO)? How were such thresholds selected? Did the authors assess whether varying the thresholds impacted the values of the quantified data? There is data out there that manual thresholding can impact values.

We prefer manual thresholding because the optimal threshold for selecting puncta varies according to the local background which is not uniform. An absolute intensity threshold does not always work very well. The human eye is good at detecting puncta from local background and the process can be made relatively objective by performing all analysis blind to the experimental group.

Reviewer #2:The authors' key conclusion is that the LRRTM1 and 2 molecules play different roles in different places at different times. This is an important conclusion, though most of the observations here are not very surprising based on the existing literature.With early and widespread knockout, synapse numbers (spines, PSDs, mini frequency) are reduced in CA1 but not dentate gyrus. This is not surprising given that we know LRRTM3/4 play important roles in DG and thus are available to compensate there, but are not expressed significantly in CA1.– It is confusing to describe the DKO mice from start to finish in the methods without reference to the previous cDKO mice. My impression is that the mice line used in Bhouri 2018 was crossed with Ella-Cre mice here. Is that correct, or were they regenerated with the same tools as the mice described in Bhouri 2018?

We apologize for the confusion. The reviewer is correct that the mice line used in Bhouri et al., 2018 was crossed with Ella-Cre mice in this study. We have revised the methods section to correctly reflect this.

– Parallel measurements in DG and CA1 would strengthen the argument that this paper provides the definitive demonstration of unique and separable roles in the two cell types. Importantly, excluding all detail from analysis of DG synapses in the cDKO animals not only provides a stark contrast with the extensive analysis in the DKO animals but will make for a difficult time interpreting the effects on LTP. Ideally, the analyses in Figure 1S2, 2, 3, and 4 would be carried out in granule cells and their perforant path synapses, though rather than repeating this battery to search for negative effects, it might be sufficient merely to provide a more specific measure of synapse number and a representative measure of synapse function.

In our manuscript, we focused on cognitive assessment because we observed reduced synaptic function as well as plasticity in dorsal hippocampus when LRRTM1/2 was deleted (both Global DKO and Local late cDKO). Hippocampus-dependent long-term contextual learning is encoded primarily by the dorsal hippocampus and its associated structures (Dong et al., 2009; Fanselow and Dong, 2010). Both the CA1 and the DG play central roles in encoding context-dependent long-term memories in the brain (Frank et al., 2004; Hainmueller, 2020; Stefanelli et al., 2016; Wiltgen et al., 2006). Therefore, our study design only allowed us to focus on the role of LRRTM1/2 in dorsal hippocampal (CA1+DG)-dependent cognitive function.

– The effect on post-tetanic potentiation following the stimulus appears strikingly different in Figure 4A and 6B, perhaps suggestive of differential presynaptic effects during the tetanic stimulus, but no assessment of presynaptic function is provided.

This was addressed in Essential revision #1.

– The LTP experiments in Figure 4 begin with an unbalanced synaptic drive. If 20% of the synapses are lacking, then delivering the stimulus at the same proportion of the maximal response will also be presumably driving the cells with 20% less input. This could cause a difference in the magnitude of LTP. The comparison thus appears a bit unfair, particularly without any assessment of presynaptic function e.g. during the train induction where the remaining DKO synapses could have more quickly decrementing release. (The authors may point out that no difference in PTP magnitude in Figure 4 indirectly argues against this possible difference in presynaptic function.)

This was addressed in Essential revision #1.

There is further no attempt at discerning a behavioral role for the LTP deficit in DG.Some clarification on staining and synapse counting would be useful. It seems that the only analysis of synapse numbers with the late cKO is VGlut or VGAT staining. Are AMPARs lost from the remaining synapses in the early or late DKO neurons?

In a related study (Karimi et al., 2021), we had shown that AMPAR levels were reduced in the mediodorsal thalamus in *Lrrtm1* cKO mice. With sparser neurons, it was technically feasible for us to assess synaptic levels of AMPAR subunits in the mediodorsal thalamus. This measure is a bit more unreliable in the hippocampus because of the higher synaptic density. In future studies, we plan to use expansion microscopy to accurately assess synaptic surface levels of AMPARs in hippocampal subregions.

Reviewer #3:Dhume and colleagues investigate the contribution of the postsynaptic adhesion molecules LRRTM1 and LRRTM2 to hippocampal circuit function. By crossing floxed LRRTM1 and LRRTM2 mice with the EIIa Cre line, they generate double knockout (DKO) mice. By broadly infecting floxed LRRTM1 and LRRTM2 mice with Cre recombinase expressing AAV, they generate conditional DKO (cDKO) mice. These strategies allow the authors to compare the effects of chronic LRRTM1/2 deletion in DKO mice vs. the effects of LRRTM1/2 deletion in cDKO mice after synapse development has largely completed. The authors find reduced excitatory synapse density in the CA1 region of LRRTM1/2 DKO mice but not cDKO mice. LTP is impaired in both DKO and cDKO mice, confirming previous results (Bhouri et al., PNAS 2018). Contextual fear conditioning is impaired in both DKO and cDKO mice.The experiments in the paper are well-performed and clearly presented, and provide additional insight in the role of LRRTM1/2 in circuit development and function. The study is a follow up to the previously published work by Bhouri et al., but uses two different deletion strategies and adds data on synapse density and behavioral paradigms. The LTP experiments presented are similar to that study, with the difference that Bhouri et al. recorded LTP in sparsely infected CA1 neurons from LRRTM1/2 cDKO mice compared to the population measurements in this study, and the additional analysis of LTP in cDKO DG in this study.1) Figure 1 —figure supplement 1 panels B and C appear to show western blot analysis of LRRTM1 and LRRTM2 levels in different LRRTM1 and LRRTM2 genotypes, but the figure legend mentions PCR. Please clarify.

We apologize for the confusion. We have corrected the figure legend to describe the western blot analysis of LRRTM1 and LRRTM2 levels.

2) If this is western blot as the text, blot and markers in kDa suggest, which LRRTM antibodies were used? I could not find this information in the Methods section but may have missed it. The bands detected that appear to represent LRRTM1 and LRRTM2 run low, at or below the predicted molecular weight of 58 kDa (uniprot) for these proteins, which seems low for glycosylated membrane proteins. The band pattern also appears different from what was reported before in Bhouri et al., PNAS 2018. Please explain.

We have described the specific anti-LRRTM antibodies in the Methods section “AntiLRRTM1 antibody (BC267) (Bhouri et al., 2018) and anti-LRRTM2 antibody (512KSCN) (Bhouri et al., 2018).” The reviewer is correct that the predicted molecular weights of mouse LRRTM1 and 2 full lengths are 58.7 kDa and 58.8 kDa, respectively. However, without their signal peptides, the molecular weights of mouse LRRTM1 and 2 are 54.9 kDa and 55.3 kDa, respectively. The band sizes in the Western blotting therefore are correct. The difference in band pattern is likely due to different sample preparation methods, boiling versus heating at 55 degrees. Our goal was to maximize the signal from the correct LRRTM band.

3) As a suggestion, the Results section might benefit from a more explicit comparison in the text between the two strategies used by the authors (i.e. chronic LRRTM1/2 deletion in Figures 1-4 vs broad viral vector-mediated deletion of LRRTM1/2 in 3 week-old animals after synapse development has completed in Figure 5-7). Now the text for the cDKO mice section emphasizes analysis in the mature brain (page 11), whereas the timepoint of analysis (injection in 3 week-old mice + 3 weeks for analysis) does not differ much from the DKO mice (6-7 weeks old). As another example, the description of the contextual fear memory experiment in Figure 7 is more elaborate than the description for (what appears to be) the same experiment in Figure 4. The findings from the two genetic strategies could be contrasted more clearly, and in some cases, such as the fear memory experiment, even be presented side-by-side for example.

We thank the reviewer for this insightful suggestion. The revised manuscript now contains results from three different deletion models (Global DKO, Local early cDKO and Local late cDKO) of LRRTM1/2 in the hippocampus rather than the previous two deletion models (Global DKO and Local late cDKO). Although, the models have many parallel measurements (discussed in Essential revision 3), we believe that presenting results from each deletion model individually would help with the flow of the revised manuscript. The results from parallel measurements are discussed more thoroughly in the Discussion section of the manuscript.

References

Bhouri, M., Morishita, W., Temkin, P., Goswami, D., Kawabe, H., Brose, N., Sudhof, T.C., Craig, A.M., Siddiqui, T.J., and Malenka, R. (2018). Deletion of LRRTM1 and LRRTM2 in adult mice impairs basal AMPA receptor transmission and LTP in hippocampal CA1 pyramidal neurons. Proc Natl Acad Sci U S A 115, E5382-E5389.

Dong, H.W., Swanson, L.W., Chen, L., Fanselow, M.S., and Toga, A.W. (2009). Genomic-anatomic evidence for distinct functional domains in hippocampal field CA1. Proceedings of the National Academy of Sciences of the United States of America 106, 11794-11799.

Fanselow, M.S., and Dong, H.W. (2010). Are the dorsal and ventral hippocampus functionally distinct structures? Neuron 65, 7-19.

Frank, L.M., Stanley, G.B., and Brown, E.N. (2004). Hippocampal plasticity across multiple days of exposure to novel environments. Journal of Neuroscience 24, 7681-7689.

Hainmueller, T. (2020). Dentate gyrus circuits for encoding , retrieval and discrimination of episodic memories. Nature Reviews Neuroscience 21, 153-168.

Karimi, B., Silwal, P., Booth, S., Padmanabhan, N., Dhume, S.H., Zhang, D., Zahra, N., Jackson, M.F., Kirouac, G.J., Ko, J.H., et al. (2021). Schizophrenia-associated LRRTM1 regulates cognitive behavior through controlling synaptic function in the mediodorsal thalamus. Molecular Psychiatry.

Stefanelli, T., Bertollini, C., Lüscher, C., Muller, D., and Mendez, P. (2016). Hippocampal Somatostatin Interneurons Control the Size of Neuronal Memory Ensembles. Neuron 89, 1074-1085.

Wiltgen, B.J., Sanders, M.J., Anagnostaras, S.G., Sage, J.R., and Fanselow, M.S. (2006). Context fear learning in the absence of the hippocampus. Journal of Neuroscience 26, 5484-5491.